# Multi-Pathway Mechanisms of Engeletin in Ischemic Stroke: A Comprehensive Study Based on Network Pharmacology, Machine Learning, and Immune Infiltration Analysis

**DOI:** 10.3390/ijms262311446

**Published:** 2025-11-26

**Authors:** Huiming Xue, Yuchen Wen, Jiahui Yang, Yue Zhang, Chang Jin, Bing Li, Yongxing Ai, Meizhu Zheng, Boge Wen, Kai Song

**Affiliations:** 1College of Life Sciences, Changchun Normal University, Changchun 130032, China; 13844637912@163.com (H.X.); 18296862081@163.com (Y.W.); yjh15043935568@168.com (J.Y.); 15942907284@163.com (Y.Z.); j1504408478@163.com (C.J.); 18544359016@163.com (B.L.); 2College of Animal Science, Jilin University, 5333 Xi’an Road, Changchun 130062, China; aiyx@jlu.edu.cn; 3Central Laboratory, Changchun Normal University, Changchun 130032, China; songkai@ccsfu.edu.cn; 4College of Computer Science & Engineering, Changchun University of Technology, Changchun 130012, China

**Keywords:** Engeletin, ischemic stroke, network pharmacology, machine learning, neuroprotection, immune profiling analysis

## Abstract

Ischemic stroke (IS) is a leading cause of mortality and long-term disability, underpinned by complex molecular mechanisms, such as oxidative stress, neuroinflammation, and apoptosis. The flavonoid Engeletin exhibits promising neuroprotective properties, but its mechanism of action remains largely unknown. In this study, we employed a systems biology approach, combined with artificial intelligence (AI), to uncover the multitarget mechanisms of Engeletin in IS. Potential targets were predicted using SwissTargetPrediction and PharmMapper and were found to intersect with IS-related genes from multiple disease databases. Functional enrichment analyses (GO/KEGG) revealed significant involvement in three classical neuroprotective pathways: PI3K-Akt-mTOR/Caspase/BCL2 (anti-apoptotic), TLR4/NF-κB (anti-inflammatory), and NRF2/KEAP1/HO-1 (antioxidant). Notably, we integrated six machine learning models (RF, SVM, GLM, KNN) to identify robust IS-specific biomarkers from the GSE22255 transcriptomic database. We used CIBERSORTx to characterize immune cell infiltration patterns in IS, revealing elevated populations of CD8^+^ T cells, M0 macrophages, and other PBMC-derived immune cells, suggesting the presence of an immunologically dynamic microenvironment. Molecular docking predicted favorable binding affinities of Engeletin to core targets (e.g., EGFR, IGF1R, KEAP1, JAK2). Finally, in vitro experiments using a Na_2_S_2_O_4_-induced PC12 cell model confirmed Engeletin’s efficacy in reducing oxidative stress, modulating calcium overload, and regulating apoptosis- and inflammation-related genes. Overall, our study establishes a comprehensive pharmacological mechanistic framework for Engeletin in combating IS and reveals the multitarget and multi-pathway neuroprotective mechanisms, thus providing preliminary support for using Engeletin in combating ischemic stroke.

## 1. Introduction

Ischemic stroke (IS) is widely recognized as one of the most systemic diseases, accounting for ~80% of all stroke cases [1]. The pathology of IS involves complex interactions between multiple biological systems, including neuronal apoptosis, peripheral immune activation, metabolic dysregulation, and neuroinflammation [2,3]. Although acute recanalization, such as intravenous thrombolysis and mechanical thrombectomy, have significantly improved short-term outcomes, their clinical benefits are constrained by limited time windows and hemorrhagic risk, meaning that they largely fail to reverse neuronal injury [4]. Therefore, neuroprotective interventions targeting subacute and chronic phases of IS are urgently needed.

Traditional stroke therapies have focused on single-target intervention, aiming to block apoptosis, inflammation, or oxidative stress individually. However, omics-driven studies and systems biology have revealed that IS is a network-level disorder characterized by multi-pathway disruption and cross-talk [5,6]. This shift in understanding calls for multitarget therapeutic strategies capable of restoring systemic homeostasis. In this context, natural compounds with multitarget, multitier regulatory features have emerged as promising candidates.

Engeletin, a naturally occurring dihydroflavonol isolated from Engelhardia roxburghiana, has a defined chemical structure (C_16_H_14_O_7_) and moderate oral bioavailability. Recent studies have shown that Engeletin exhibits anti-inflammatory, antioxidant, and anti-apoptotic effects across various disease models [7,8]. Engeletin (CAS 572-31-6) is commercially available from suppliers such as Sigma-Aldrich, TCI, and BioCrick (HPLC ≥ 98%, ~¥400/20 mg) as a single enantiomer with consistent optical rotation (−16° to −18°). Notably, it protects the vascular endothelium by regulating the PI3K/Akt/eNOS signaling pathway, suppresses TLR4/NF-κB activation to alleviate inflammatory lung injury, and enhances NRF2/HO-1-mediated antioxidant defense to improve diabetic nephropathy [9,10,11]. Other dihydroflavonols, such as dihydroquercetin, have been reported to provide neuroprotection in subarachnoid hemorrhage by activating the PI3K/AKT/Nrf2/HO-1 pathway and inhibiting ferroptosis, suggesting that this subclass of flavonoids may share a common mechanistic basis for combating neurological disorders [12]. Similarly, baicalein, a flavonoid from Scutellaria baicalensis, has recently been shown to alleviate cerebral ischemia–reperfusion injury by regulating the GPX4/ACSL4/ACSL3 axis to suppress ferroptosis [13]. These core pathways also play pivotal roles in the pathogenesis of ischemic stroke [14,15], suggesting that Engeletin may have therapeutic potential for combating stroke. However, the exact mechanisms of action of Engeletin remain to be fully elucidated.

To address this, our study integrates systems pharmacology with machine learning and immune deconvolution to systematically decode Engeletin’s therapeutic mechanisms in IS. Initially, we identified several Engeletin targets using SwissTargetPrediction and PharmMapper that overlap with IS-related genes derived from GeneCards, OMIM, PharmGKB, and DisGeNET databases. Functional enrichment analyses (GO/KEGG) revealed significant involvement in PI3K-Akt-mTOR/Caspase/BCL2, TLR4/NF-κB, and NRF2/KEAP1/HO-1 pathways. To refine key targets, we applied six machine learning algorithms (RF, SVM, GLM, KNN) to gene expression profiles from the GSE22255 database, that identified IS-associated markers. Immune infiltration analysis using CIBERSORTx uncovered significant changes in CD8^+^ T cells, M0 macrophages, and neutrophils, thus highlighting the role of immunopathology in IS.

Mechanistically, these three pathways do not function independently, but instead form a complex signaling interaction network during the pathophysiological process of stroke. PI3K/Akt can inhibit NF-κB activation by suppressing IKK activity, thereby reducing the spread of inflammation [16]. There exists a transcriptional competition between NF-κB and NRF2, wherein the former can suppress NRF2 activity by sequestering CBP, thereby weakening antioxidant defense [17]. Activation of NRF2 not only alleviates oxidative stress, but it also inhibits inflammation and apoptosis by downregulating the TLR4/NF-κB pathway through HO-1 [18]. As a downstream pathway of PI3K/Akt, mTOR is not only involved in metabolic regulation and autophagy, but it also enhances NRF2 activity through the p62-KEAP1 axis. Moreover, it plays a crucial role in regulating macrophage polarization that contributes to alleviating post-stroke immune imbalance [19,20].

Molecular docking verified strong binding affinities of Engeletin to targets, such as Nrf2, Keap1, HO-1, PI3K, and AKT. Finally, in vitro experiments using the Na_2_S_2_O_4_-induced PC12 ischemia model confirmed its efficacy in modulating oxidative stress, calcium overload, and key apoptotic/inflammatory gene expression. Together, our study builds a multi-layered mechanistic framework of Engeletin against IS, offering valuable insights into its translational potential as a multitarget neuroprotective agent (Figure 1).

## 2. Results

### 2.1. Network Pharmacology Prediction and Enrichment Analysis of Engeletin in IS

By using SwissTargetPrediction and PharmMapper, we found that 4539 ischemic stroke (IS)-related targets intersected, and GeneCards, OMIM, PharmGKB, and DisGeNET identified 197 overlapping genes (Figure 2a). A protein–protein interaction (PPI) network was constructed using the STRING database (Figure 2b), and identified 10 topologically important hub genes: GFR, HSP90AA1, SRC, ESR1, PTPN11, JAK2, MET, PTK2, KDR, and IGF1R. These were identified using the cytoHubba plugin in Cytoscape (Figure 2c). Gene Ontology (GO) enrichment analysis showed that these targets are primarily involved in biological processes, such as oxidative stress, inflammation, and apoptosis (Figure 2d). Kyoto Encyclopedia of Genes and Genomes (KEGG) pathway analysis indicated significant enrichment in several pathways, including PI3K-Akt signaling, MAPK signaling, Ras signaling, EGFR tyrosine kinase inhibitor resistance, and fluid shear stress and atherosclerosis (Figure 2e).

Although pathways labeled under cancer (e.g., “prostate cancer”, “proteoglycans in cancer”, “endocrine resistance”) were also enriched, we concluded that this is likely due to shared regulatory mechanisms involved in cellular proliferation, inflammatory modulation, and oxidative stress response, which are also relevant to IS pathology. Further construction of a pathway–gene interaction network revealed that several core genes—EGFR, IGF1R, SRC, CASP3, JAK2, and MAP2K1—participate in multiple interconnected signaling pathways (Figure 2f). These genes may function as topologically important nodes, rather than being interpreted as direct causal regulators. For example, EGFR serves as a converging node among the Ras, PI3K-Akt, and MAPK pathways, which regulate apoptosis, while IGF1R acts as an upstream regulator of the PI3K-Akt axis, involved in insulin signaling and neuronal repair. CASP3, as an executor of apoptosis, is activated downstream of multiple signaling cascades. Collectively, these results suggest that Engeletin may exert neuroprotective effects against ischemic stroke via a multitarget and multi-pathway regulatory mechanism, involving the modulation of apoptosis, inflammation, and oxidative stress.

### 2.2. Multi-Model Machine Learning Prediction and Enrichment Analysis

Based on the GEO GSE22255 dataset, we constructed predictive models using four machine learning algorithms—Generalized Linear Model (GLM), K-Nearest Neighbors (KNN), Random Forest (RF), and Support Vector Machine (SVM)—and applied the permutation importance method to identify key predictive genes (Table 1 and Figure 3a). To prevent model overfitting, we employed Bootstrap resampling and five-fold cross-validation for robustness evaluation. The results demonstrated that the SVM model performed the best, achieving an accuracy of 58.33%, an AUC-ROC of 0.67, and an F1 score of 0.615 on the test set, showing a good balance between sensitivity and specificity (Figure 3b,c). In contrast, the RF model showed lower accuracy (41.67%, AUC-ROC = 0.50), the GLM achieved similar accuracy to SVM (58.33%) but had a poor model fit, and the KNN model performed moderately (AUC-ROC = 0.53). Bootstrap validation further confirmed that the SVM model maintained the highest stability across repeated sampling (AUC = 0.669), while the other models exhibited greater variability and weaker robustness (Figure 3e). Further predictive analysis indicated that the SVM model could accurately distinguish between “stroke” and “control” samples, with only a few borderline cases (e.g., GSM554015 and GSM554033) showing probabilities close to 0.5, reflecting minor uncertainty due to the small sample size. Overall, the SVM model demonstrated superior classification accuracy, generalization ability, and stability, making it suitable for gene screening and early diagnosis of ischemic stroke.

To explore its biological significance, Figure 3d shows the feature importance distribution of the top 15 key genes identified by the SVM model, among which PLA2G2A, EGFR, RARα, CAT, NRF2, and ADORA1 contributed most significantly. These features are closely associated with the PI3K-Akt-mTOR/BCL2 (anti-apoptotic), TLR4/NF-κB (anti-inflammatory), and NRF2/KEAP1/HO-1 (antioxidant) signaling pathways, indicating a strong alignment between the model-identified features and disease-related molecular mechanisms.

To validate the consistency between the machine learning results and network pharmacology predictions, we further performed GO and KEGG enrichment analyses on the key genes identified by the SVM model. These analyses were conducted using the entire human genome as the background/reference set to ensure standardized and unbiased identification of over-represented biological terms. The results (Figure 3f) showed that these genes were primarily involved in nutrient response, inflammatory regulation, ERK/MAPK cascade, and lipid metabolism, localized mainly in the plasma membrane, vesicles, and endoplasmic reticulum, and exhibited nuclear receptor activity, transcription factor binding, and phosphatase binding functions. Combined with the network pharmacology findings, these genes were functionally consistent with the PI3K-Akt-mTOR/BCL2 (anti-apoptotic), TLR4/NF-κB (anti-inflammatory), and NRF2/KEAP1/HO-1 (antioxidant) signaling pathways, revealing a dynamic regulatory mechanism among metabolic stress, inflammatory response, and oxidative balance. The three pathways intersect through MAPK, AMPK, HIF-1, and FoxO signaling to form an “anti-apoptotic–anti-inflammatory–antioxidant” regulatory network that jointly determines cellular survival, repair, or death. Dysregulation of this network is closely related to the development of cancer, metabolic syndrome, and chronic inflammatory diseases, making it a crucial target for multi-pathway drug intervention and disease prevention.

Table 2 lists the top ten KEGG pathways identified from the SVM model’s top 15 key genes, which show the highest relevance to the three major signaling systems predicted through network pharmacology: PI3K-Akt-mTOR/Caspase/BCL2 (anti-apoptotic), TLR4/NF-κB (anti-inflammatory), and NRF2/KEAP1/HO-1 (antioxidant). The pathways were enriched against the background of the whole human genome. The enriched pathways, including PI3K-Akt, MAPK, HIF-1, Ras, and NF-κB signaling, are functionally involved in regulating cell proliferation, inflammation, apoptosis inhibition, and oxidative stress defense.

These results indicate that the machine-learning-based analysis of the GSE22255 dataset and the network pharmacology predictions converge on the same mechanistic network—comprising metabolic regulation, inflammation control, and redox homeostasis—thereby mutually validating each other. This consistency strengthens the overall findings, suggesting that the compounds or active ingredients under study may exert therapeutic effects through coordinated modulation of the PI3K-Akt/NF-κB/NRF2 signaling axis, forming an integrated “anti-apoptotic–anti-inflammatory–antioxidant” defense network at the molecular level.

### 2.3. Immune Infiltration Analysis Based on Clinical Data

To elucidate potential immunological roles of Engeletin-targeted genes in the context of IS, we performed immune cell composition analysis based on the GSE22255 dataset from the GEO database using the CIBERSORTx algorithm. This approach allowed us to estimate the relative abundance of 22 immune cell subsets in peripheral blood mononuclear cell (PBMC) samples from IS patients and healthy controls (Figure 4). Notably, among the 22 immune cell types provided by CIBERSORTx, some cell types (such as neutrophils) are not typical components of PBMCs. Their estimated values may originate from cross-hybridization or contamination of transcriptomic signals. Therefore, we strictly limit the interpretation scope herein to typical PBMC subsets, including T cells, B cells, monocytes, NK cells, among others. All comparative analyses were corrected for the False Discovery Rate (FDR), with a significance threshold set at FDR < 0.05. This analysis revealed a notable elevation of activated mast cells in the IS group (effect size = 0.03998), representing the most prominently upregulated immune subset. This finding suggests that mast cells contribute to IS-related inflammation and/or modulation of vascular permeability. Additionally, both resting and activated subtypes of CD4 memory T cells were significantly increased (effect sizes = 0.01442 and 0.00654, respectively), indicating a potential involvement of sustained or secondary adaptive immune responses in the pathophysiology of IS. In contrast, a marked reduction was observed in resting natural killer (NK) cells (effect size = −0.02799), the most strongly downregulated cell type, thus implying compromised innate immune function or NK cell exhaustion. Monocyte levels were also significantly decreased (effect size = −0.01734), which may be associated with post-stroke immunosuppression or peripheral-to-central immune cell trafficking (Figure 5).

Collectively, these findings delineate a biphasic immunological profile in IS patients characterized by enhanced adaptive immune activation (mast cells, CD4 memory T cells) and suppressed innate immune responses (NK cells, monocytes). This imbalance may underlie increased susceptibility to secondary infections and provides a novel perspective on post-stroke immune dysregulation that offers potential avenues for immunomodulatory strategies. Further integration with network pharmacology and machine learning analyses suggests that the observed depletion of monocytes and NK cells may be attributable to increased immune cell apoptosis mediated by dysregulation of the PI3K-Akt signaling pathway. Concurrently, the activation of mast cells and CD4 memory T cells is likely driven by upregulated pro-inflammatory signaling via the TLR4/NF-κB axis. Moreover, the substantial reduction in monocytes may reflect attenuation of NRF2/KEAP1/HO-1 antioxidant defense mechanisms, thereby weakening their cytoprotective capacity. These findings indicate that PI3K-Akt, NF-κB, and NRF2 signaling pathways may form an intricate regulatory network that shapes the post-stroke immune microenvironment, thus highlighting their potential as targets for therapeutic immunomodulation.

### 2.4. Molecular Docking with Predicted Pathway Proteins

To further assess the binding potential of Engeletin to key stroke-related targets, molecular docking was performed using 11 core proteins involved in relevant signaling pathways. Binding affinities (kcal/mol) and interaction types were evaluated. As shown in Table 3, the docking results predicted that Engeletin could bind to all target proteins, with binding energies lower than −7.0 kcal/mol, indicating favorable affinity. Among them, Keap1 (PDB ID: 1U6D) showed the strongest binding affinity (−11.7 kcal/mol), forming multiple hydrogen bonds (e.g., ILE559A, VAL561A) and hydrophobic interactions. This in silico finding suggests that Engeletin might have the potential to disrupt Keap1–Nrf2 binding, which could, in turn, promote Nrf2 release and activation of antioxidant responses. Nrf2 itself also was predicted to have a strong binding affinity (−9.9 kcal/mol), aligning with its central role in redox regulation.

In the apoptosis pathway, Engeletin formed stable interactions with both Bcl-2 (−8.5 kcal/mol) and BAX (−8.3 kcal/mol), including multiple hydrogen bonds and hydrophobic contacts. Notably, Bcl-2 formed several hydrogen bonds with Engeletin (e.g., TRP195A, TYR9A), implying the potential for modulating apoptotic balance. For cell survival signaling, Engeletin demonstrated good binding to AKT (−7.5 kcal/mol), PI3K (−8.0 kcal/mol), and mTOR (−7.6 kcal/mol) that involve formation of hydrogen bonds, hydrophobic interactions, and π–π stacking. This supports its likely regulation of apoptosis inhibition and neuroprotection via the PI3K/Akt/mTOR axis. In terms of inflammation, Engeletin showed stable binding to TLR4 (−8.6 kcal/mol) and NF-κB (−7.1 kcal/mol), forming salt bridges (e.g., ARG106D, ARG2305A) and hydrogen bonds, suggesting a potential role in attenuating stroke-related inflammation via the TLR4/NF-κB pathway. Additionally, stable binding with effector molecules such as HO-1 (−8.0 kcal/mol) and Caspase-3 (−7.1 kcal/mol) provides preliminary in silico evidence that Engeletin might exert multitarget, multi-pathway regulatory effects in the context of ischemic stroke (Figure 6).

### 2.5. Neuroprotective Effect of Engeletin in the OGD/R Model

#### 2.5.1. Morphological Changes in PC12 Cells

Bright-field microscopy showed marked cellular damage in the OGD/R group, including reduced cell number, irregular morphology, and evidence of cell rounding and detachment, indicating successful model establishment. Treatment with Edaravone (15 μM) partially ameliorated these changes. Notably, Engeletin exerted a concentration-dependent protective effect, with the 25 μM group exhibiting near-normal morphology, high cell density, intact structure, and good adherence (Figure 7).

#### 2.5.2. Comprehensive Protective Effects of Engeletin in the OGD/R Model

OGD/R treatment significantly reduced cell viability (Figure 8a) and mitochondrial membrane potential (Figure 8b), indicating functional impairment and mitochondrial instability. Concurrently, we observed increased LDH release (Figure 8c) and elevated ROS levels (Figure 8d). Engeletin treatment ameliorated these effects in a dose-dependent manner, with the 25 μM group showing the most pronounced protection that restored cell viability and mitochondrial function, as well as reducing LDH and ROS levels. These effects were comparable to those seen with the positive control Edaravone. Flow cytometry (Figure 8e) further confirmed that OGD/R induced substantial apoptosis, with a significantly increased rate of apoptosis (Figure 8f). Engeletin treatment effectively suppressed apoptosis, with the 25 μM group reducing the apoptosis rate to ~14%, comparable to the Edaravone group (Figure 8).

#### 2.5.3. Engeletin Attenuates OGD/R-Induced Mitocghondrial Membrane Potential Loss

JC-1 staining revealed a significant reduction in mitochondrial membrane potential in the OGD/R group, as indicated by increased green fluorescence. Treatment with Engeletin progressively restored membrane potential in a dose-dependent manner, as evidenced by enhanced red fluorescence. The 25 μM group exhibited the most pronounced recovery, comparable to that of the Edaravone group (Figure 9a). As shown in Figure 9b, the OGD/R model group exhibited a marked decline in the JC-1 red/green fluorescence ratio, indicating a significant loss of mitochondrial membrane potential (## *p* < 0.01 vs. control). Treatment with Engeletin effectively reversed this reduction in a dose-dependent manner, as evidenced by a gradual increase in the red/green fluorescence ratio across the 6.25, 12.5, and 25 μM groups (*p* < 0.01 vs. model). Among them, the 25 μM Engeletin group showed the most pronounced recovery of mitochondrial potential, reaching a level comparable to that of the positive control Edaravone (15 μM). These results suggest that Engeletin can significantly mitigate OGD/R-induced mitochondrial dysfunction and maintain mitochondrial membrane integrity, thereby exerting a neuroprotective effect against ischemia–reperfusion injury.

### 2.6. Experimental Validation of Predicted Target Proteins

#### 2.6.1. Engeletin Activates the PI3K/Akt/mTOR Signaling Pathway Under OGD/R Conditions

Western blotting showed that exposure to OGD/R markedly reduced expression of phosphorylated PI3K, AKT, and mTOR, indicating suppression of the PI3K/Akt/mTOR pathway. Engeletin treatment significantly upregulated the expression of these phosphorylated proteins in a concentration-dependent manner. The 25 μM group demonstrated the strongest activation, approaching levels observed in the Edaravone-treated group. These findings suggest that Engeletin exerts neuroprotective effects by activating the PI3K/Akt/mTOR pathway (Figure 10).

#### 2.6.2. Engeletin Inhibits OGD/R-Induced Mitochondrial-Pathway-Mediated Apoptosis

OGD/R stimulation led to apoptotic activation, characterized by increased expression of cleaved caspase-3 and a decreased Bcl-2/Bax ratio. Engeletin treatment reduced cleaved caspase-3 expression and restored the Bcl-2/Bax ratio in a dose-dependent manner, with the 25 μM group showing the most substantial improvement, closely matching that of the Edaravone group. These results indicate that Engeletin effectively inhibits mitochondria-dependent apoptosis under OGD/R conditions (Figure 11).

#### 2.6.3. Engeletin Inhibits the TLR4/NF-κB Inflammatory Signaling Pathway Under OGD/R Conditions

Western blotting showed that OGD/R significantly increased expression of TLR4 and phosphorylated NF-κB p65, indicating activation of the pro-inflammatory TLR4/NF-κB axis. Engeletin treatment downregulated both TLR4 and p-NF-κB expression in a dose-dependent manner. Notably, the 25 μM group showed expression levels comparable to those in the Edaravone group, suggesting that Engeletin may exert anti-inflammatory effects by suppressing the TLR4/NF-κB pathway (Figure 12).

#### 2.6.4. Engeletin Activates the NRF2/KEAP1/HO-1 Antioxidant Pathway Under OGD/R Conditions

Exposure to OGD/R suppressed the antioxidant response, as characterized by reduced nuclear NRF2 accumulation and HO-1 expression, alongside elevated levels of KEAP1. Engeletin treatment reversed these changes by promoting NRF2 nuclear translocation and HO-1 upregulation, while reducing KEAP1 expression. The 25 μM group exhibited effects comparable to Edaravone, suggesting that Engeletin enhances cellular antioxidant capacity via activation of the NRF2/KEAP1/HO-1 pathway (Figure 13).

## 3. Discussion

In this study, we systematically evaluated neuroprotective effects of Engeletin in an ischemic stroke (IS) model by integrating network pharmacology, machine learning, immune infiltration analysis, molecular docking, and in vitro validation. Our integrated approach suggests that Engeletin exerts its therapeutic effects via a “multitarget, multi-pathway” regulatory mechanism that is focused on three major signaling cascades implicated in IS pathophysiology: PI3K-Akt-mTOR/Caspase/BCL2 [21,22,23], TLR4/NF-κB [17,24,25], and NRF2/KEAP1/HO-1 [26,27,28].

Multi-database integration identified 197 overlapping targets between Engeletin and IS. PPI network analysis revealed hub proteins that included EGFR, JAK2, SRC, and CASP3. GO and KEGG enrichment indicated significant involvement of targets in classical stroke-associated pathways, such as PI3K-Akt, MAPK, JAK-STAT, and AGE-RAGE [29]. Machine learning algorithms applied to the GSE22255 dataset consistently highlighted CASP3, BCL2L1 (apoptosis), JAK2, STAT1 (inflammation), and GSR, HSPA8 (oxidative stress), confirming the biological plausibility of the predicted network [30].

Based on CIBERSORTx immune infiltration analysis of typical PBMC subsets, IS patients exhibited elevated adaptive immune cells (activated mast cells, CD4 memory T cells) alongside decreased innate immune cells (monocytes, resting NK cells), reflecting an “adaptive activation–innate suppression” imbalance [31]. This profile is likely linked to TLR4/NF-κB pathway hyperactivation and PI3K/Akt signaling suppression, thus highlighting these pathways as potential immunomodulatory targets of Engeletin. Molecular docking revealed that Engeletin binds strongly to several key proteins—Keap1 (−11.7 kcal/mol), Nrf2 (−9.9 kcal/mol), Bcl-2 (−8.5 kcal/mol), TLR4 (−8.6 kcal/mol), primarily by forming hydrogen bonds, hydrophobic interactions, π-stacking, and salt bridges, thus offering a structural basis for its multi-pathway regulation [32].

In vitro, Engeletin (especially at 25 μM) significantly restored cell viability, improved mitochondrial membrane potential (ΔΨm), and reduced LDH release and ROS accumulation [23,33]. Flow cytometry and Western blotting showed that Engeletin attenuated mitochondrial apoptosis by decreasing cleaved Caspase-3 and restoring the Bcl-2/Bax ratio a It also suppressed TLR4 and p-NF-κB expression, while enhancing NRF2 nuclear translocation, increasing HO-1 expression, and decreasing KEAP1 levels [17,28], with effects comparable to those of Edaravone.

Importantly, these three pathways are interlinked. PI3K/Akt activation enhances NRF2 stability and transcriptional activity [26], which strengthens antioxidative capacity and indirectly inhibits the caspase cascade. NRF2-induced HO-1 not only neutralized ROS, but also inhibited TLR4/NF-κB-mediated inflammation [28]. Conversely, NF-κB hyperactivation suppressed NRF2 and Bcl-2 expression, thereby promoting oxidative stress and apoptosis [17,27]. Thus, Engeletin’s multitarget regulatory action may re-establish redox and immune balance, which may represent an integrated neuroprotective mechanism of action.

PI3K–Akt–mTOR, TLR4/NF-κB, and NRF2/KEAP1/HO-1 are key signaling pathways involved in regulating cell survival, inflammation, and oxidative stress [17,26,27,28]. The PI3K–Akt–mTOR pathway is activated by various cell stimuli, such as growth factors. PI3K catalyzes production of PIP3 that recruits and activates Akt (aka protein kinase B). Akt then phosphorylates downstream targets, such as mTOR, GSK3β, and BAD to promote protein synthesis, inhibit autophagy, and maintain cell survival and proliferation by suppressing pro-apoptotic proteins (such as BAD and Caspase-9) and enhancing anti-apoptotic proteins (such as Bcl-2) expression [34,35]. Akt also indirectly activates the NF-κB pathway, thereby enhancing its role in cell protection [36].

The TLR4/NF-κB pathway is a key inflammatory pathway in the innate immune system [34,35,36,37]. When TLR4 recognizes pathogen-associated molecular patterns, such as lipopolysaccharides (LPS), it recruits MyD88 or TRIF adapter proteins to activate the IKK complex that leads to degradation of IκB and release of NF-κB (p65/p50) to translocate to the nucleus and induce expression of pro-inflammatory factors, like IL-6 and TNF-α, thereby initiating an inflammatory response [37,38]. In addition, NF-κB regulates expression of genes related to apoptosis, cell cycle, and immune regulation, making it a crucial transcriptional regulatory hub under stress conditions [39].

The NRF2/KEAP1/HO-1 pathway is the primary cellular antioxidant defense mechanism [40,41,42]. In the resting state, NRF2 is captured by KEAP1 and directed for ubiquitination and degradation. When cells face oxidative stress, critical cysteine residues undergo conformational changes, allowing NRF2 to dissociate from KEAP1 and translocate to the nucleus. There, it binds to antioxidant response elements (AREs) and induces expression of antioxidant genes, such as HO-1, NQO1, SOD, and CAT, thereby alleviating oxidative stress and maintaining cellular homeostasis [38,39]. NRF2 also negatively regulates the NF-κB pathway, thereby establishing a dynamic balance between inflammation and oxidative stress [42].

Here, we have established an Endeletin-based mechanistic framework with immune infiltration characteristics, integrating immune profiling, target prediction, and molecular docking to connect immune status recognition with drug regulation. Our study revealed that Engeletin reverses the post-stroke immune pattern of “adaptive activation and innate suppression,” thereby extending the understanding of its pharmacological effects beyond the traditionally known antioxidant and anti-apoptotic activities. In addition, we clarified the coordinated regulatory mechanisms among the TLR4/NF-κB, NRF2/HO-1, and PI3K/Akt signaling pathways, linking them to changes in specific immune cell subsets, thereby enhancing the interpretability and translational potential of the findings. By potentially modulating multiple immune-related pathways, Engeletin may regulate the activation and homeostasis of key immune cells, suggesting a multi-dimensional intervention in the post-stroke immune microenvironment and its associated anti-inflammatory, antioxidant, and neuroprotective effects. Overall, our study highlights the multi-faceted therapeutic potential of Engeletin against IS. Future studies are warranted to explore single-cell transcriptomic profiling, specific target gene knockouts (e.g., Keap1, TLR4, AKT), pharmacokinetic optimizations, and combination therapies with thrombolytics or anti-platelet agents (Figure 14).

This study primarily investigated the mechanism of action of Engeletin through an integrated approach of computational simulations and in vitro experiments. Although these results reveal its potential to act via the PI3K/Akt, NRF2/HO-1, and TLR4/NF-κB pathways, it is important to note that in vivo efficacy validation in animal models has not yet been conducted, as we were constrained by the current research stage and resources. Consequently, conclusions regarding its therapeutic efficacy require final confirmation through future in vivo experiments. Based on the solid mechanistic evidence provided herein, subsequent in vivo studies have been planned as a key focus for our next steps.

## 4. Materials and Methods

### 4.1. Target Identification and Network Pharmacology Analysis

To elucidate neuroprotective mechanisms of Engeletin against IS, a network pharmacology approach was employed. The structure of Engeletin was retrieved from PubChem (National Center for Biotechnology Information, Bethesda, MD, USA) and used for target prediction via SwissTargetPrediction (Swiss Institute of Bioinformatics, Lausanne, Switzerland) and PharmMapper (version 2017, Shanghai Institute of Materia Medica, Shanghai, China) (species: *Homo sapiens*) [43]. IS-related genes were obtained from GeneCards (Weizmann Institute of Science, Rehovot, Israel), OMIM (Johns Hopkins University, Baltimore, MD, USA), PharmGKB (Stanford University, Palo Alto, CA, USA), and DisGeNET (Universitat Politècnica de València, Valencia, Spain) using “Ischemic stroke” as the keyword [44]. Targets were standardized and deduplicated through UniProt (Swiss Institute of Bioinformatics, Lausanne, Switzerland). Overlapping targets were identified via Venny (version 2.1, Centro de Investigación Príncipe Felipe, Valencia, Spain) and input into the STRING database (version 11.5, EMBL, Heidelberg, Germany; confidence ≥ 0.7) for PPI network construction [45]. Cytoscape (v3.8.0, Institute for Systems Biology, Seattle, WA, USA) and its cytoHubba plugin (v0.1, Institute for Systems Biology, Seattle, WA, USA) were used to screen hub genes. GO and KEGG enrichment analyses were performed using the DAVID database (version 2021, Frederick National Laboratory for Cancer Research, Frederick, MD, USA; *p* ≤ 0.01), using the whole human genome as the background and setting the significance threshold at FDR < 0.05 adjusted by the Benjamini–Hochberg (BH) method with visualizations generated via Cytoscape and online bioinformatics tools [46].

### 4.2. Multi-Model Machine-Learning-Based Target Prioritization

This study was based on the publicly available transcriptomic dataset GSE22255 from the Gene Expression Omnibus (GEO) database, which includes 40 peripheral blood mononuclear cell (PBMC) samples—20 from ischemic stroke (IS) patients and 20 from healthy controls. All data were generated using the Affymetrix Human Genome U133 Plus 2.0 Array (GPL570) platform [47]. To explore the potential molecular targets of Engeletin, core differential genes were first selected from the GSE22255 expression matrix as feature variables to construct four machine learning classification models: Generalized Linear Model (GLM), K-Nearest Neighbors (KNN), Random Forest (RF), and Support Vector Machine (SVM), with the sample category (IS vs. Control) as the response variable [48]. Key feature genes were identified using the Permutation Importance method, and both Bootstrap resampling (1000 iterations) and five-fold cross-validation (5-fold CV) were employed to prevent overfitting. Model performance was evaluated using AUC-ROC, AUC-PR, Accuracy, Sensitivity, Specificity, Precision, and F1-score. The results indicated that the SVM model demonstrated the best predictive performance (AUC = 0.67, F1 = 0.615), the RF model performed poorly (AUC = 0.50), the GLM model showed signs of overfitting (Pseudo R^2^ = −12.8), and the KNN model exhibited moderate performance (AUC = 0.53). After Bootstrap validation, the SVM model maintained stable performance across resampling, making it the optimal classification model in this study. Subsequently, the DAVID and ClusterProfiler R packages (Version 4.4.1) were used to perform functional enrichment analyses on the core genes, including Gene Ontology (GO) biological process (BP), cellular component (CC), and molecular function (MF), as well as KEGG pathway analysis [49].

### 4.3. Immune Infiltration Analysis

Relative abundances of immune cell subsets in the GSE22255 dataset were deconvoluted using the CIBERSORTx algorithm [31]. The output includes estimated fractions for 22 immune cell types. However, consistent with the known composition of peripheral blood mononuclear cells (PBMCs), subsequent analysis excluded cell types such as neutrophils that are not typical PBMC constituents. Consequently, all comparative analyses and interpretations focused solely on standard PBMC-derived populations between the IS and non-IS control groups [50]. All *p*-values from group comparisons were adjusted using the False Discovery Rate (FDR) method, with a significance threshold set at FDR < 0.05. Immune cell subsets demonstrating significant alterations were then integrated with the core signaling pathways previously predicted by our network pharmacology and machine learning framework to establish potential mechanistic links [51].

### 4.4. Molecular Docking Analysis

To evaluate binding of Engeletin to stroke-related targets, molecular docking was conducted. The 3D structure of Engeletin (.sdf) was obtained from PubChem via TCMSP [52]. Crystal structures of 11 key proteins from the RCSB Protein Data Bank (PDB) (e.g., Keap1, Nrf2, PI3K, AKT, mTOR, Caspase-3, TLR4, NF-κB) and preprocessed with PyMOL (Version 3.0.1) to remove heteroatoms [53]. Docking was performed on CB-Dock2 with five repetitions per protein. Conformations with the lowest binding energies (AutoDock (Version 1.5.7) Vina score) were selected. Binding interactions were analyzed using Discovery Studio 2025, and 2D diagrams were generated to highlight hydrogen bonds, hydrophobic contacts, π–π stacking, and salt bridges [54]. This provided structural support for Engeletin’s potential in modulating oxidative stress, inflammation, and apoptosis pathways.

### 4.5. In Vitro Experimental Procedures

#### 4.5.1. Reagents and Antibodies

Engeletin (≥98% pure) and the positive control Edaravone (EDA) were purchased from YuanYe Biotechnology Co., Ltd. (Shanghai, China). High-glucose DMEM, fetal bovine serum (FBS), and penicillin–streptomycin (PS) for cell culture were obtained from Sangon Biotech (Shanghai, China). Other reagents used for evaluating oxidative stress, mitochondrial function, and apoptosis included Tris, glycine, sodium dodecyl sulfate (SDS), Earle’s Balanced Salt Solution (EBSS), lactate dehydrogenase (LDH) assay kit, Fluo-3/AM calcium fluorescent probe, MTT [3-(4,5)-dimethylthiazol-2-yl)-2,5-diphenyltetrazolium bromide], Annexin V-FITC/PI apoptosis detection kit, DCFH-DA reactive oxygen species (ROS) probe, JC-1 mitochondrial membrane potential assay kit, polyvinylidene fluoride (PVDF) membranes, SDS-PAGE gel preparation kit, and dimethyl sulfoxide (DMSO). Unless otherwise specified, all were purchased from Sigma-Aldrich (St. Louis, MO, USA). Primary antibodies used for Western blotting included β-actin, PI3K, phospho-PI3K (p-PI3K), AKT, phospho-AKT (p-AKT), mTOR, phospho-mTOR (p-mTOR), Caspase-3, Cleaved-Caspase-3, Bcl-2, Bax, KEAP1, and HO-1, all sourced from Abcam (Cambridge, UK). Antibodies against TLR4, NF-κB p65, and nuclear NRF2 were obtained from Abclonal (Wuhan, China). Horseradish peroxidase (HRP)-conjugated secondary antibodies (anti-rabbit and anti-mouse IgG) used for chemiluminescent detection were purchased from Abcam.

#### 4.5.2. In Vitro Evaluation of Engeletin’s Neuroprotective Effects

PC12 cells (purchased from Shanghai, China) were cultured in high-glucose DMEM supplemented with 10% fetal bovine serum (FBS) and 1% penicillin-streptomycin (PS) under standard conditions (37 °C, 5% CO_2_). Upon reaching 70–80% confluency, cells were trypsinized and seeded in appropriate plates for subsequent assays. The OGD/R model was induced by exposing cells to glucose-free Earle’s Balanced Salt Solution (EBSS) containing 10 mmol/L sodium dithionite (Na_2_S_2_O_4_) for 2 h, followed by 24 h reoxygenation in normal culture medium [55]. Cells were divided into the following groups: Control (untreated), OGD/R only, OGD/R + Engeletin (6.25, 12.5, 25 μM), and OGD/R + Edaravone (15 μM, positive control). Treatments were administered 1 h before OGD and maintained during reoxygenation.

#### 4.5.3. Morphological Observation

Subsequent to treatments, PC12 cells were maintained at 37 °C for 24 h. Morphological alterations in the designated groups were then evaluated using an inverted microscope (*n* = 6).

#### 4.5.4. MTT Method to Measure Cellular Activity

Cell viability was assessed using the MTT assay (Beyotime Biotechno1ogy, Haimen, China). After 48 h cultivation, cells from each group were incubated in growth media supplemented with 5 mg/mL MTT at 37 °C for 4 h. The resulting formazan crystals were dissolved in 150 mL dimethyl sulfoxide for 10 min, and absorbance was read at 490 nm on a microplate reader (Spectra Max M5, Molecular Devices, Sunnyvale, CA, USA). Cell viability was expressed as a percentage of the control.

#### 4.5.5. LDH Release Assay

A commercial LDH assay kit (Beyotime Biotechno1ogy, Haimen, China) was used to measure release of LDH from cells with damaged plasma membranes. Analysis was performed according to the manufacturer’s instructions. Briefly, after the various treatments, the plate containing cells was gently shaken and centrifuged at 250× *g* for 5 min. Culture media were transferred to the corresponding wells of another 96-well plate, the LDH reaction mix was added, and the plate was incubated for 30 min at room temperature. Finally, the absorbance was read at 450 nm using a microplate reader.

#### 4.5.6. Measurement of [Ca^2+^]i

The [Ca^2+^]i was determined as previously described [27,28]. Briefly, after exposure to each of the six treatments, cells were collected and incubated with complete medium containing 5 μM Fura-3/AM (Beyotime Biotechno1ogy, Haimen, China) at 37 °C for 45 min. Subsequently, cells were washed and re-suspended in cold balanced PBS containing 0.2% bovine serum albumin. Cells were then incubated at 37 °C for another 5 min, and the Ca^2+^ concentration was determined using a fluorescence spectrophotometer (Spectra Max M5, Molecular Devices, Sunnyvale, CA, USA) by monitoring emission changes at 510 nm with an excitation switch from 340 to 380 nm. The [Ca^2+^]i was expressed as a percentage of that from untreated control cells.

#### 4.5.7. Reactive Oxygen Species (ROS) Level Detection

Quantification of intracellular ROS was done using a fluorescent probe assay with DCFH-DA (ROS Assay Kit, Beyotime, Shanghai, China). The culture medium was first aspirated, and cells were subsequently incubated in the dark with 10 µM DCFH-DA working solution for 40 min at 37 °C. Thereafter, cells were rinsed three times with phosphate-buffered saline (PBS) to eliminate excess probe. Fluorescence intensity was determined on the multifunctional enzyme marker (λ_ex = 488 nm, λ_em = 525 nm), and resultant ROS levels were presented as relative fluorescence units.

#### 4.5.8. Measurement of Mitochondrial Membrane Potential (MMP)

PC12 cells were seeded into six-well plates at a density of 2.0 × 10^5^ cells per well and cultured for 24 h. Following the same drug treatment protocol as described above, cells underwent a 24 h reperfusion period. Mitochondrial membrane potential (MMP) was assessed using the fluorescent probe JC-1 (C2006, Beyotime Biotechnology), with incubation performed at 37 °C for 20 min in the dark. After washing with JC-1 staining buffer (1×) and trypsinization, an equal number of cells were transferred to a black 96-well plate. Fluorescence intensity was measured using a microplate reader, with excitation at 488 nm and emission collected at 590 nm (red) and 515 nm (green). MMP levels were expressed as the ratio of red to green fluorescence intensity.

#### 4.5.9. Apoptosis Rate Detection

Apoptosis was evaluated using an Annexin V-FITC/PI detection kit (BD Biosciences, San Jose, CA, USA). Cells harvested from each group were resuspended at a density of 1 × 10^6^ cells/mL. A 100 µL aliquot of the cell suspension was stained with 5 µL Annexin V-FITC and 5 µL propidium iodide (PI) in the dark for 15 min. Samples were then analyzed on a BD FACSCalibur flow cytometer. The percentages of early apoptotic (Annexin V^+^/PI^−^) and late apoptotic (Annexin V^+^/PI^+^) cells were determined using FlowJo software (version 10.8.1) [56].

### 4.6. Western Blotting

The cell lysate was prepared by adding 1 mM PMSF as the protease inhibitor at a volume ratio of 1:99 relative to the lysate. Lysis was performed on ice for ~20 min, followed by centrifugation at 4 °C and 13,000× *g* for 5 min. The resulting supernatant was collected for subsequent procedures.

After electrophoresis and transfer, membranes were incubated overnight at 4 °C with primary antibodies specific to p-Akt, t-Akt, t-PI3K, p-PI3K, BDNF, p-CREB, t-CREB, and β-actin, all diluted at 1:1000. Following washing, the blots were incubated for 45 min with peroxidase-conjugated secondary antibodies. Protein detection was carried out using an enhanced chemiluminescence (ECL) detection system. Band intensities were quantified and normalized to β-actin using the Quantity One software (v4.6.8, Bio-Rad Laboratories, Hercules, CA, USA).

### 4.7. Statistics Analysis

All data are shown as the mean ± SD of at least three independent experiments. Statistical comparisons between two groups were performed using Student’s *t*-test, whereas multiple group comparisons were performed using the one-way ANOVA followed by Dunnett’s post hoc test, using GraphPad Prism 6.0 (GraphPad Software). A *p*-value of <0.05 was considered statistically significant.

## Figures and Tables

**Figure 1 ijms-26-11446-f001:**
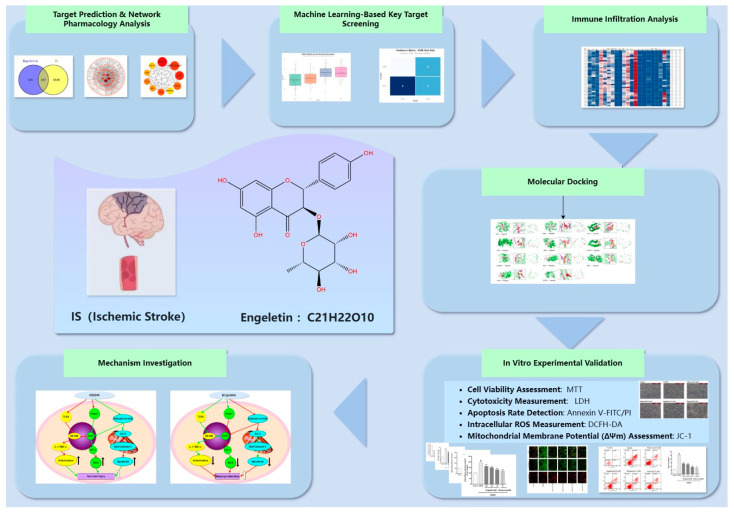
Workflow of the study: The flowchart summarizes the integrative strategy for evaluating Engeletin (C_21_H_22_O_10_) in ischemic stroke (IS), including target prediction, machine-learning-based screening, immune infiltration analysis, molecular docking, mechanistic exploration, and in vitro experimental validation.

**Figure 2 ijms-26-11446-f002:**
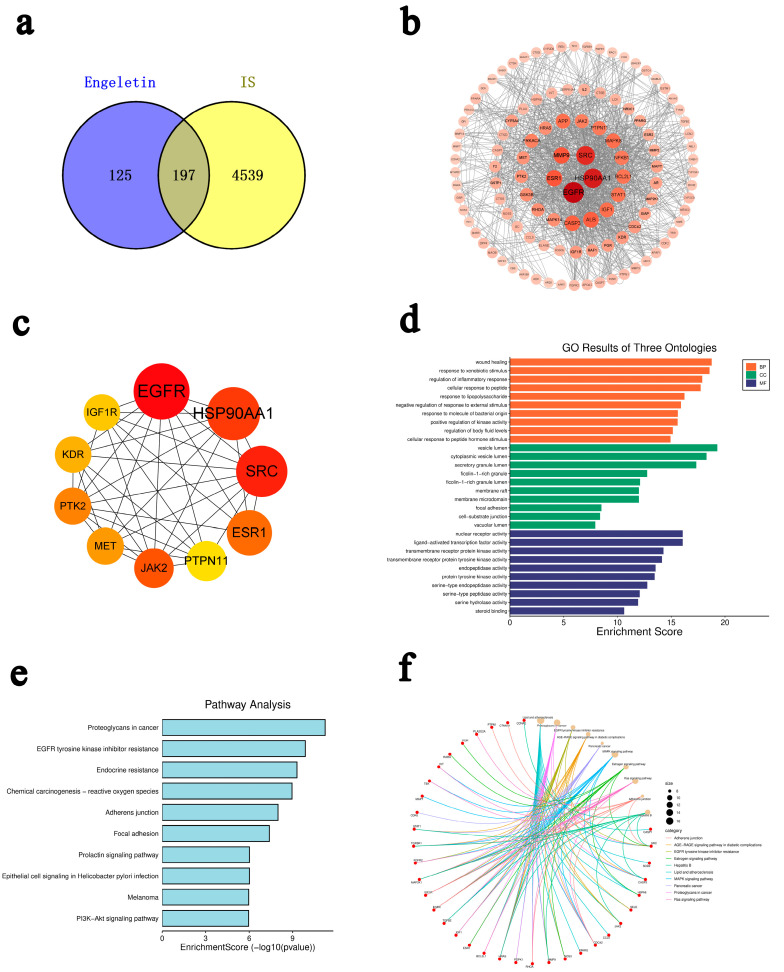
Target identification and enrichment analysis of Engeletin against ischemic stroke. (**a**) Venn diagram of overlapping drug and disease targets (197 common targets); (**b**) PPI network of common targets; (**c**) Top 10 hub genes identified by MCC algorithm; (**d**) GO enrichment (top 10 BP in green, CC in blue, MF in orange); (**e**) KEGG enrichment (top 20 pathways); (**f**) pathway–gene interaction network.

**Figure 3 ijms-26-11446-f003:**
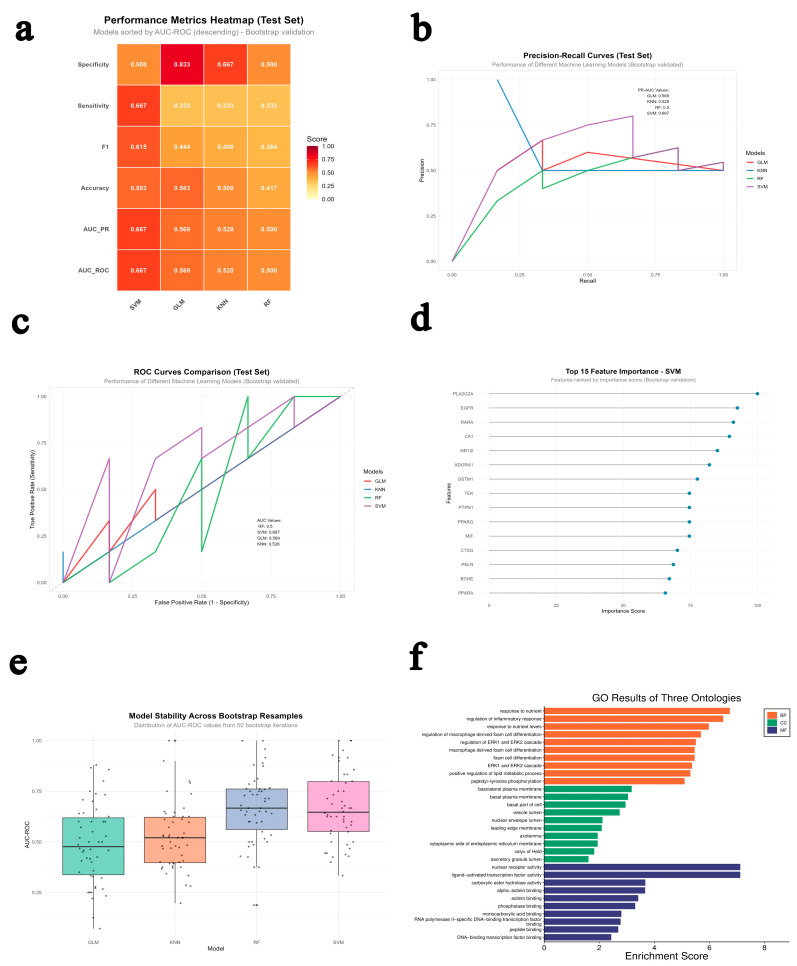
Integrated results of machine learning and bioinformatics analyses. (**a**) Performance heatmap. (**b**) Precision–recall curve. (**c**) ROC curve. (**d**) Feature importance analysis. (**e**) Bootstrap resampling validation. (**f**) Summary of top GO terms across BP, CC, and MF.

**Figure 4 ijms-26-11446-f004:**
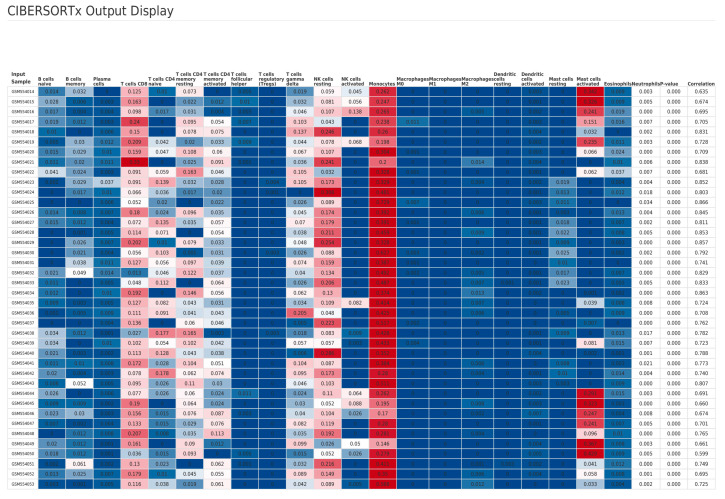
Immune cell fractions estimated by CIBERSORTx. (Heatmap of 22 immune cell types across samples.) Color Legend: The color gradient from blue to red represents the scale of immune cell infiltration levels, where blue indicates the lowest relative proportion and red indicates the highest relative proportion.

**Figure 5 ijms-26-11446-f005:**
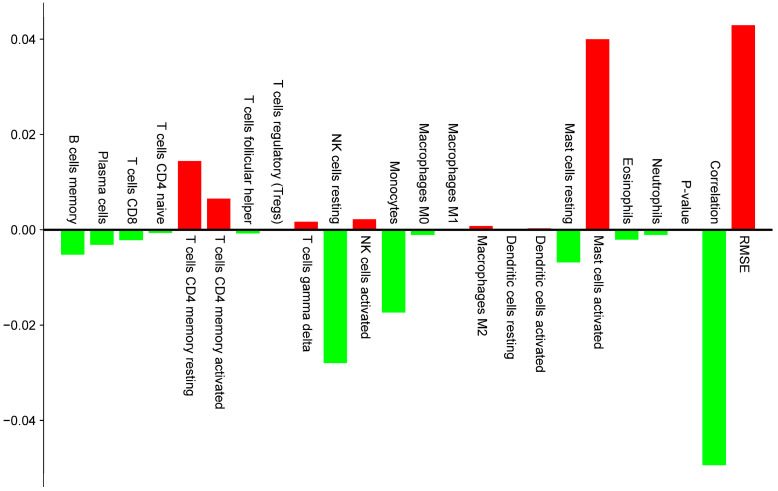
Differential analysis of immune cell infiltration. (Bar plot showing increased (red) or decreased (green) immune cell types between groups.)

**Figure 6 ijms-26-11446-f006:**
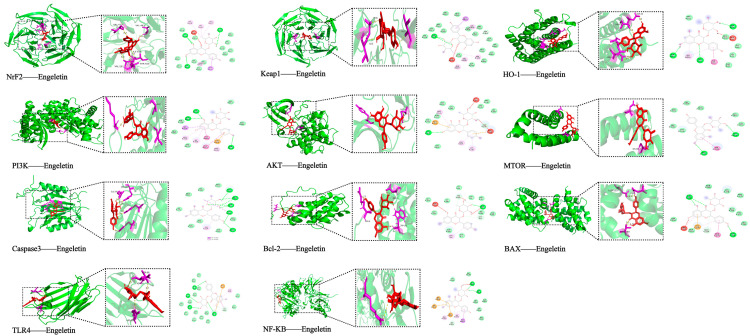
Molecular docking of Engeletin with key target proteins.Red molecule: Engeletin. Purple regions: Key residues of the target protein that interact with Engeletin. (Docking models and interaction details of Engeletin with Nrf2, Keap1, HO-1, PI3K, AKT, MTOR, Caspase-3, Bcl-2, BAX, TLR4, and NF-κB, showing binding conformations and key residues involved.)

**Figure 7 ijms-26-11446-f007:**
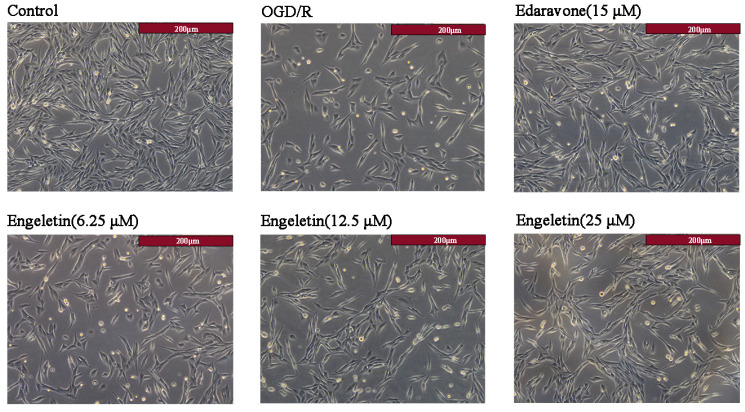
Effects of Engeletin on PC12 cell morphology under OGD/R.

**Figure 8 ijms-26-11446-f008:**
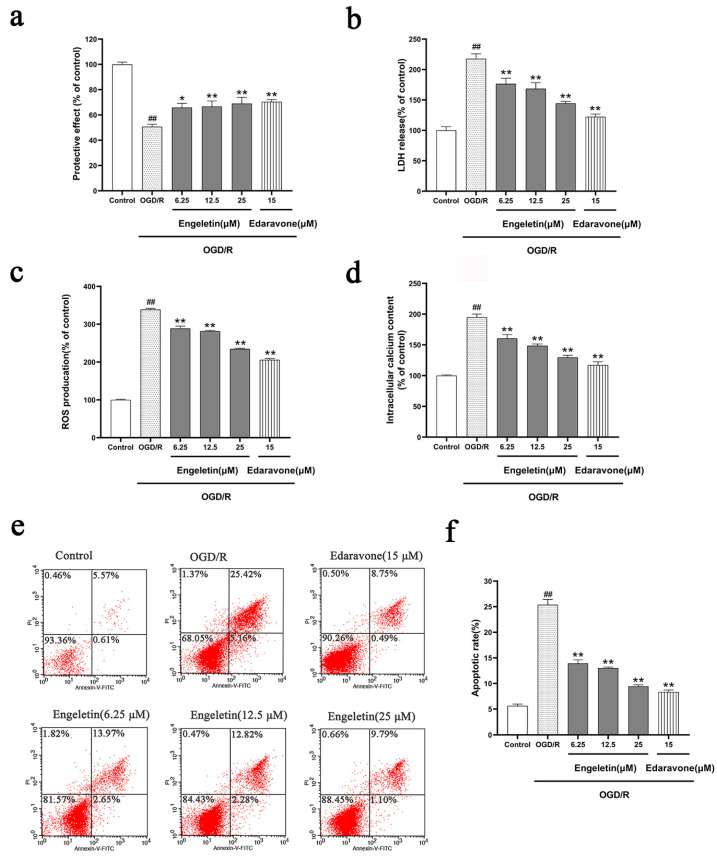
Engeletin mitigates OGD/R-induced oxidative stress, mitochondrial dysfunction, and apoptosis in vitro. (**a**,**b**) OGD/R significantly reduced cell viability and mitochondrial membrane potential (ΔΨm), both of which were dose-dependently restored by Engeletin, especially at 25 μM. (**c**,**d**) Elevated LDH release and ROS levels under OGD/R were reduced by Engeletin and Edaravone. (**e**,**f**) Flow cytometry showed increased apoptosis after OGD/R, which was markedly reversed by Engeletin in a concentration-dependent manner. (Values are indicated as mean ± SD; statistical analysis was performed using one-way ANOVA followed by Tukey’s post hoc test. *n* = 3, ## *p* < 0.01, compared to control, * *p* < 0.05, ** *p* < 0.01 versus model cohort.)

**Figure 9 ijms-26-11446-f009:**
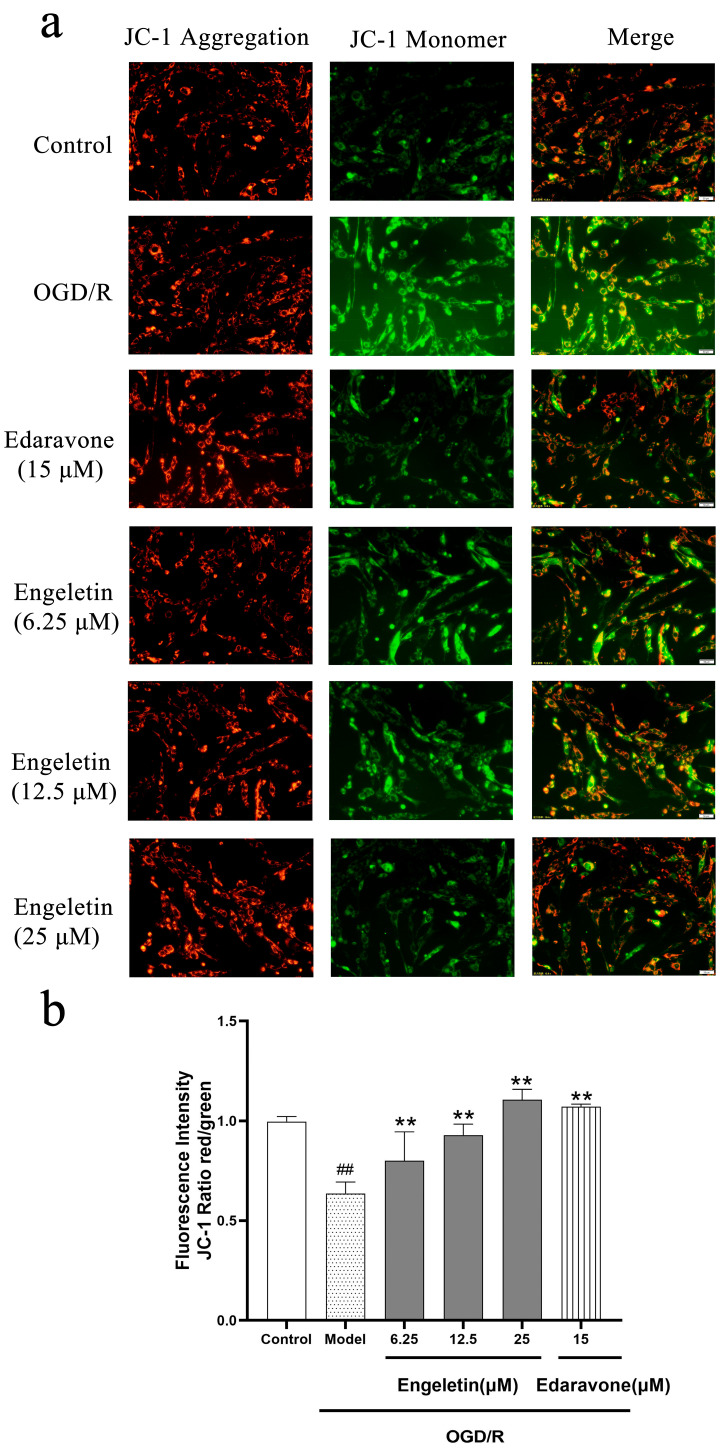
Engeletin restores mitochondrial membrane potential disrupted by OGD/R in vitro. (**a**) Representative JC-1 staining results (flow cytometry plots and merged fluorescence images). Red fluorescence indicates JC-1 aggregates (high ΔΨm), while green fluorescence indicates monomers (low ΔΨm). OGD/R treatment (Model) shifted fluorescence from red to green, which was reversed by Engeletin and Edaravone treatments. (A scale bar of 50 μm is included in all fluorescence images). (**b**) Quantitative analysis of the JC-1 red/green fluorescence intensity ratio. (Values are indicated as mean ± SD; statistical analysis was performed using one-way ANOVA followed by Tukey’s post hoc test. *n* = 3, ## *p* < 0.01, compared to control, ** *p* < 0.01 versus model cohort.)

**Figure 10 ijms-26-11446-f010:**
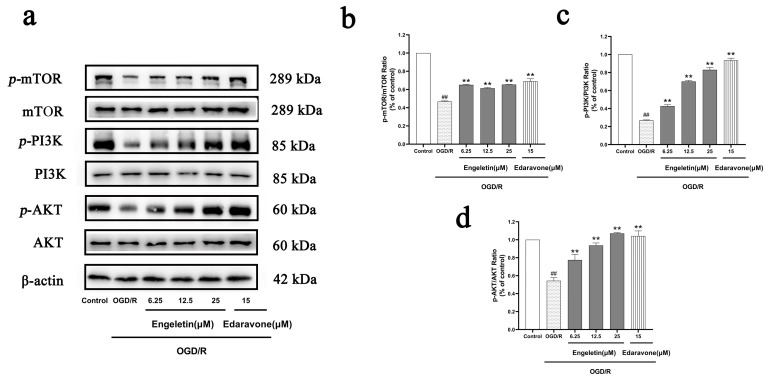
Engeletin activates the PI3K/Akt/mTOR signaling pathway under OGD/R conditions. (**a**) Western blot analysis of p-PI3K, p-AKT, and p-mTOR and their total protein expression levels; (**b**–**d**) quantification of phosphorylation ratios (p-PI3K/PI3K, p-AKT/AKT, and p-mTOR/mTOR). (Values are indicated as mean ± SD; statistical analysis was performed using one-way ANOVA followed by Tukey’s post hoc test. *n* = 3, ## *p* < 0.01, compared to control, ** *p* < 0.01 versus model cohort.)

**Figure 11 ijms-26-11446-f011:**
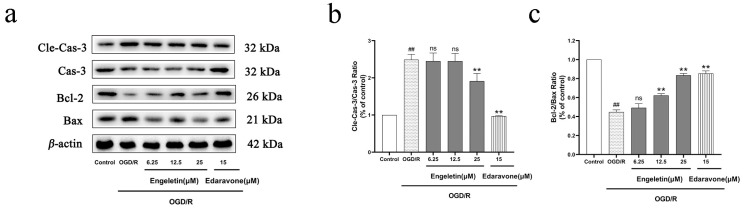
Engeletin suppresses mitochondrial apoptosis-related protein expression under OGD/R conditions. (**a**) Western blot analysis of Caspase-3, Cleaved-Caspase-3, Bcl-2, and Bax protein levels in each group; (**b**) quantification of the Cleaved-Caspase-3/Caspase-3 ratio; (**c**) Bcl-2/Bax ratio analysis. (Values are indicated as mean ± SD; statistical analysis was performed using one-way ANOVA followed by Tukey’s post hoc test. *n* = 3, ## *p* < 0.01, compared to control, ** *p* < 0.01 versus model cohort, ns = not significant.)

**Figure 12 ijms-26-11446-f012:**
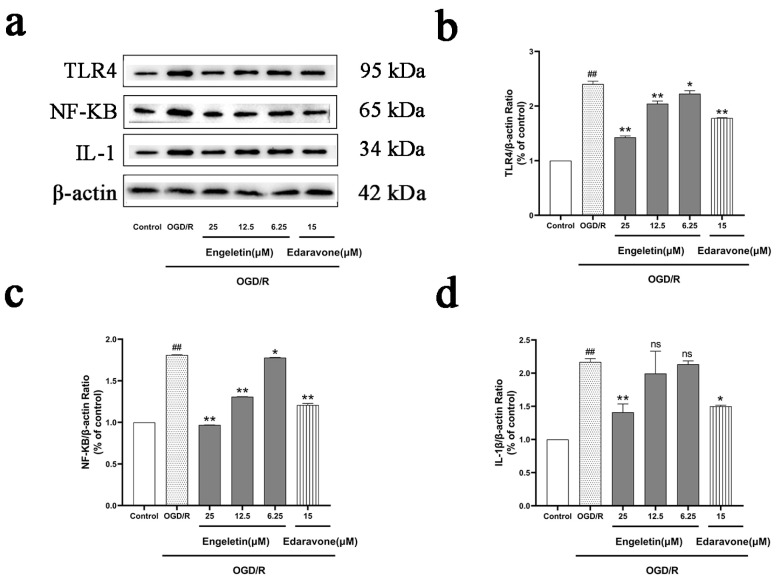
Engeletin inhibits the TLR4/NF-κB signaling pathway under OGD/R conditions. (**a**) Western blot analysis of TLR4, IL-1 and p-NF-κB p65; (**b**) Quantitative analysis of TLR4, normalized to β-actin; (**c**) Quantitative analysis of the p-NF-κB p65/NF-κB p65 ratio; (**d**) Relative quantification of IL-1 expression. (Values are indicated as mean ± SD; statistical analysis was performed using one-way ANOVA followed by Tukey’s post hoc test. *n* = 3, ## *p* < 0.01, compared to control, * *p* < 0.05, ** *p* < 0.01 versus model cohort, ns = not significant.)

**Figure 13 ijms-26-11446-f013:**
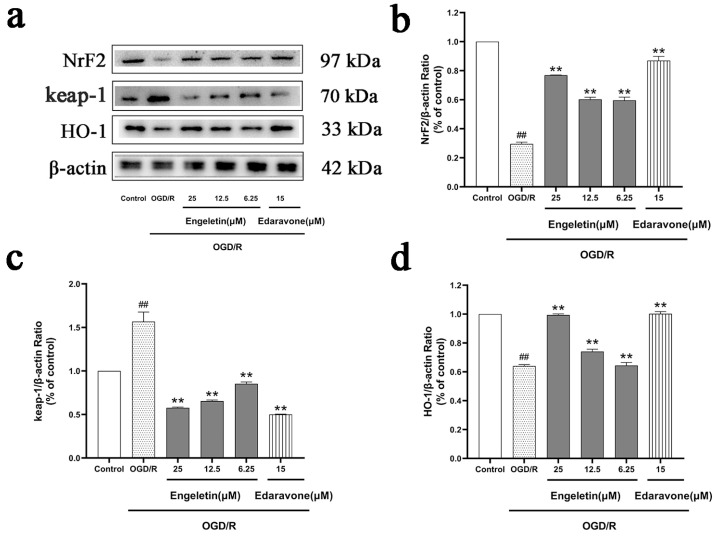
Engeletin activates the NRF2/KEAP1/HO-1 antioxidant pathway under OGD/R conditions. (**a**) Western blot analysis of NRF2 (nuclear), KEAP1, and HO-1; (**b**–**d**) Quantification of NRF2/β-actin, KEAP1/β-actin, and HO-1/β-actin. Engeletin enhanced antioxidant signaling in a concentration-dependent manner. (Values are indicated as mean ± SD; statistical analysis was performed using one-way ANOVA followed by Tukey’s post hoc test. *n* = 3, ## *p* < 0.01, compared to control, ** *p* < 0.01 versus model cohort.)

**Figure 14 ijms-26-11446-f014:**
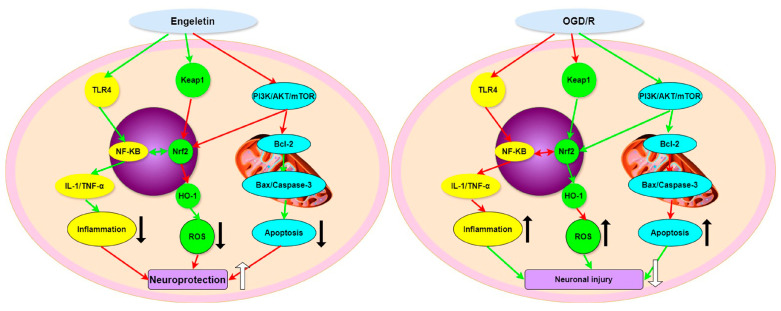
Mechanism of Engeletin in ischemic stroke. Engeletin protects neurons by activating Nrf2/HO-1 and PI3K/AKT/mTOR pathways and inhibiting NF-κB, thereby reducing inflammation, ROS, and apoptosis.Red arrows indicate an inhibitory effect. Green arrows indicate an activating or promoting effect.

**Table 1 ijms-26-11446-t001:** Comparative performance of four machine learning models (RF, SVM, GLM, KNN) on the test set.

Model	Data_Type	Accuracy	Sensitivity	Specificity	Precision	F1	AUC_ROC	AUC_PR
RF	Test	0.4167	0.3333	0.5000	0.4000	0.3636	0.5000	0.5000
SVM	Test	0.5833	0.6667	0.5000	0.5714	0.6154	0.6667	0.6667
GLM	Test	0.5833	0.3333	0.8333	0.6667	0.4444	0.5694	0.5694
KNN	Test	0.5000	0.3333	0.6667	0.5000	0.4000	0.5278	0.5278

**Table 2 ijms-26-11446-t002:** Top 10 KEGG pathways related to the PI3K-Akt/NF-κB/NRF2 signaling axes identified from SVM model analysis.

Rank	Pathway	KEGG ID	Main Associated System	Related Function Overview
1	PI3K-Akt signaling pathway	hsa04151	PI3K-Akt-mTOR/BCL2	It primarily regulates cell growth, metabolism, and anti-apoptosis by activating Akt to upregulate mTOR and BCL2 signaling.
2	MAPK signaling pathway	hsa04010	PI3K-Akt/NF-κB	MAPK can cross-activate the NF-κB and mTOR pathways, regulating cell inflammation and survival.
3	HIF-1 signaling pathway	hsa04066	PI3K-Akt/NRF2	Closely related to oxidative stress, stabilizes HIF-1α through PI3K/Akt and promotes antioxidant responses.
4	Ras signaling pathway	hsa04014	PI3K-Akt/NF-κB	Activation of PI3K-Akt and MAPK jointly promotes growth and anti-apoptosis.
5	NF-κB signaling (via TLR4/NF-κB crosslink)	(hsa04620/04151)	TLR4/NF-κB	Core pathways of inflammatory response, regulating the expression of immune factors and cell survival.
6	JAK-STAT signaling pathway	hsa04630	NF-κB/PI3K-Akt	Promotes the expression of anti-inflammatory cytokines while interacting with NF-κB.
7	AMPK signaling pathway	hsa04152	NRF2/KEAP1/PI3K-Akt	Can activate NRF2 and inhibit mTOR, thereby enhancing antioxidant capacity and metabolic homeostasis.
8	FoxO signaling pathway	hsa04068	PI3K-Akt/NRF2	Akt inhibits FoxO transcription factors, which regulate the expression of antioxidant enzymes (such as SOD and CAT).
9	TNF/Toll-like receptor signaling pathway	hsa04620	TLR4/NF-κB	A key upstream regulator of inflammation and cell death, linking immunity and oxidative stress responses.
10	Glutathione metabolism	hsa00480	NRF2/KEAP1/HO-1	The main antioxidant metabolic pathways regulated by NRF2 maintain cellular redox balance.

**Table 3 ijms-26-11446-t003:** Docking results of Engeletin with key target proteins, including protein data, binding affinity, RMSD, and major interaction types (hydrophobic, hydrogen bonds, salt bridges, π-stacking).

Protein	PDB ID	Ligands	Affinity (kcal/mol)	RMSD	Bonding Interaction
AKT	1GZK	engeletin	−7.5	0.605 Å	Hydrophobic Interactions (LYS 216A, GLU230A, GLU433A) Hydroge nBondsInte Ractions (ARG208A, ASN233A, GLU433A, LYS285A) Salt Bridges (LYS285A, LYS290A)
BAX	BAX	engeletin	−8.3	0.607 Å	Hydrophobic Interactions (ARG22C, GLN221C, LYS276A, PHE273A, THR280A) Hydrogen Bonds Interactions (GLN229C, LYS276A, THR280A)
Bcl-2	1G5M	engeletin	−8.5	1.625 Å	Hydrophobic Interaction (ARG6A, ASN11A) Hydrogen Bond Interactions (TRP195A, ASN11A, ASP10A, TYR9A, ARG6A)
Caspase-3	1CP3	engeletin	−7.1	0.610 Å	Hydrophobic Interaction (TYR329B, VAL390B) Hydrogen Bond Interactions (ARG286A, LYS259A, TYR331B)
foxo	3CO6	engeletin	−8.1	0.603 Å	Hydrophobic Interaction (VAL1188C, TRP1206C) Hydrogen Bond Interactions (GLU1185C, GLY1198C, LYS1195C) Salt Bridges (LYS1207C)
Hif1	1H2K	engeletin	−7.1	0.613 Å	Hydrophobic Interactions (PHE114A) Hydrogen Bond Interactions (ARG251A, LEU101A, LYS115A, TYR230A)
HO-1	1N3U	engeletin	−8.0	0.609 Å	Hydrophobic Interactions (PHE33A, ASN36A, ASP45A, LEU21A) Hydrogen Bond Interactions (ASN36A, ASP45A)
IL-1	2ILA	engeletin	−8.5	0.616 Å	Hydrophobic Interactions (GLU11B) Hydrogen Bond Interactions (ARG9B, GLU10B, LYS27A) Salt Bridges (ARG9B)
keap1	1U6D	engeletin	−11.7	0.610 Å	Hydrophobic Interactions (VAL467A) Hydrogen Bond Interactions (ILE559A, VAL420A, VAL465A, VAL561A, VAL608A)
MTOR	4DRI	engeletin	−7.6	0.611 Å	Hydrogen Bond Interactions (ARG111A, ASN92A, ILE61A, LYS87A, TYR58A) π-Stacking (PHE209A)
NF-KB	1MDI	engeletin	−7.1	0.608 Å	Hydrophobic Interactions (PRO2307A, LYS2309A) Hydrogen Bond Interactions (ARG2043A, LYS2039A) Salt Bridges (ARG2305A)
NQ01	1D4A	engeletin	−9.8	0.605 Å	Hydrophobic Interactions (PRO102B GLU117D Hydrogen Bond Interactions (TRP105B, PHE106B, PHE120D)
nrf2	2FLU	engeletin	−9.9	0.609 Å	Hydrophobic Interactions (VAL420B, ILE559B) Hydrogen Bond Interactions (ILE559B, THR560B, VAL420B, VAL465B)
PI3K	3IHY	engeletin	−8.0	0.606 Å	Hydrophobic Interactions (ILE634B, ILE760B, LEU750B) Hydrogen Bond Interactions (LYS613B, SER614B)
TLR4	2Z62	engeletin	−8.6	0.604 Å	Hydrogen Bond Interactions (SER98D) Salt Bridges (ARG106D)

## Data Availability

The data that support the findings of this study are available within the article. The transcriptomic data used in this study were obtained from the GEO database (Accession Number: GSE22255, https://www.ncbi.nlm.nih.gov/geo/, accessed on 8 June 2025). CIBERSORTx analysis was performed via the online platform at https://cibersortx.stanford.edu/ (accessed on 4 June 2025). Molecular docking was conducted using the CB-Dock2 tool at https://cadd.labshare.cn/cb-dock2/php/index.php (accessed on 15 October 2025). The scripts and parameter files used for building the machine learning model have been uploaded to the Appendix A.

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
