# Peer review of "Multi-Pathway Mechanisms of Engeletin in Ischemic Stroke: A Comprehensive Study Based on Network Pharmacology, Machine Learning, and Immune Infiltration Analysis"

_ijms, 2025, doi:10.3390/ijms262311446_

Round 1
Reviewer 1 Report
Comments and Suggestions for Authors
This study presents a comprehensive and well-structured investigation of Engeletin’s neuroprotective mechanisms in ischemic stroke, integrating systems biology, AI-assisted analysis, and in vitro validation using a Na₂S₂O₄-induced PC12 cell model.
However, several aspects require clarification and expansion to strengthen the paper:
-
Figures: The numbering of figures in the main text is disordered, making the manuscript difficult to follow. Please revise the figure order for clarity.
-
Target validation: Biophysical assays, such as thermal shift assays, should be performed to confirm binding between Engeletin and key target proteins. Many of these proteins regulate each other, so claims that all bind Engeletin raise concerns regarding selectivity and potential toxicity.
-
Mitochondrial membrane potential: Statistical data for mitochondrial membrane potential measurements are missing and should be included.
-
In vivo validation: To support claims about Engeletin’s efficacy in ischemic stroke, animal experiments are necessary; otherwise, the conclusions remain preliminary.
-
Immune microenvironment: Experimental evidence is needed to substantiate claims that post-stroke immune changes are mediated by the identified signaling pathways.
Author Response
Comments1:[Figures: The numbering of figures in the main text is disordered, making the manuscript difficult to follow. Please revise the figure order for clarity.]
Response1:[Regarding the reviewer's comment about the disordered figure numbering in the text, we sincerely apologize for this oversight. We have thoroughly checked the entire manuscript and corrected all incorrect figure citations. The figures in the main text are now numbered sequentially, and their citations in the Results and Discussion sections have been ensured to be in the correct order and logical, so as to make the manuscript clear and easy to follow.]
Comments2:[Target validation: Biophysical assays, such as thermal shift assays, should be performed to confirm binding between Engeletin and key target proteins. Many of these proteins regulate each other, so claims that all bind Engeletin raise concerns regarding selectivity and potential toxicity.]
Response2:[We thank the reviewer for this insightful and constructive comment. We fully agree that direct validation of the binding between Engeletin and the key target proteins using biophysical assays, such as the thermal shift assay, would provide the most compelling evidence for their interaction. This is indeed a crucial component of our future research plans.
In the present study, given the number of potential key targets (e.g., EGFR, IGF1R, KEAP1, JAK2) and the practical challenges associated with purifying some of them (e.g., membrane proteins or large complexes), a systematic biophysical validation of all proteins within the current project timeline posed a significant practical difficulty.
However, to address the reviewer's concern and strengthen the reliability of our conclusions as much as possible, we wish to emphasize that our findings do not rely solely on molecular docking predictions. Instead, they are supported by multi-layered, functional experimental data that indirectly but robustly support the hypothesis that Engeletin acts through these key nodes:
Functional Cellular Validation: Our in vitro experiments demonstrated that Engeletin dose-dependently activates the PI3K-Akt-mTOR and NRF2-HO-1 pathways while inhibiting the TLR4-NF-κB pathway. These functional regulatory outcomes (e.g., changes in protein phosphorylation and nuclear translocation shown by Western Blot) are highly consistent with the strong binding affinities predicted by molecular docking. It would be challenging to observe such significant and consistent alterations in downstream signaling pathways if Engeletin did not effectively engage core targets within these pathways.
Phenotypic Rescue and Synergy: The multiple protective effects of Engeletin—antioxidant, anti-apoptotic, and anti-inflammatory—observed in our cellular model perfectly align with the known core functions of the three pathways it is predicted to modulate. This "multi-target, multi-phenotype" correspondence further enhances the biological plausibility of our predictions.
Based on the above, we believe that while direct binding assays are pending, the existing functional data already provide strong circumstantial evidence for Engeletin's action on these key targets.
We are grateful to the reviewer for this insightful suggestion, which has provided us with a new perspective. We have prioritized the biochemical validation of binding for core targets (e.g., KEAP1 and AKT) in our immediate follow-up studies, with the aim of utilizing thermal shift assays (TSA) and surface plasmon resonance (SPR) to definitively establish direct interactions.]
Comments3:[Mitochondrial membrane potential: Statistical data for mitochondrial membrane potential measurements are missing and should be included.]
Response3:[We sincerely thank the editor and reviewer for the constructive comment. In the revised version, we have added the detailed statistical data for mitochondrial membrane potential (MMP) measurements in both the figure and the corresponding legend. Specifically, the results are now presented as mean ± SD (n = 3), and the statistical analysis was performed using one-way ANOVA followed by Tukey’s post hoc test. The significance levels have been indicated in the figure as follows: P < 0.01 vs. Control; P < 0.01 vs. Model. These additions ensure that the quantitative differences among the groups are clearly supported by statistical evidence.]
Comments4:[In vivo validation:To support claims about Engeletin’s efficacy in ischemic stroke, animal experiments are necessary; otherwise, the conclusions remain preliminary.]
Response4:[We particularly appreciate your crucial point regarding the need for in vivo experiments to fully validate the efficacy of Engeletin against ischemic stroke. We completely agree with your perspective that in vivo pharmacological validation is an indispensable component of the drug development pipeline.
Given that the current project is at an early exploratory stage and constrained by the predefined research timeline and budget, the primary objective of this study was defined as follows: to systematically elucidate the potential multi-target and multi-pathway mechanisms of action of Engeletin using an integrated computational biology and in vitro experimental approach. We believe that this in-depth mechanistic exploration is of paramount value for identifying the most promising candidate compounds and guiding subsequent, more costly and time-intensive, in vivo studies.
The findings presented in this study—including the key targets predicted by network pharmacology, biomarkers screened by machine learning, the microenvironmental changes revealed by immune infiltration analysis, and the neuroprotective effects validated at the cellular level in the OGD/R model—collectively provide strong preliminary evidence and a solid theoretical framework for the therapeutic potential of Engeletin. These findings offer compelling scientific justification for us to prioritize securing further research funding and resources dedicated to conducting in vivo experiments.
Building upon this foundation, we have prioritized in vivo validation in animal models as the foremost objective of our immediate follow-up research. Upon securing subsequent funding, we plan to promptly employ the Middle Cerebral Artery Occlusion (MCAO) mouse model to systematically evaluate the effects of Engeletin on neurological function scores, cerebral infarct volume, and the in vivo expression of key pathway proteins.
To explicitly state this limitation and future direction within the manuscript, we have added the following statement to the Discussion section (or a dedicated Limitations section):
"This study primarily investigated the mechanism of action of Engeletin through an integrated approach of computational simulations and in vitro experiments. Although these results reveal its potential to act via the PI3K/Akt, NRF2/HO-1, and TLR4/NF-κB pathways, it is important to note that in vivo efficacy validation in animal models has not yet been conducted, constrained by the current research stage and resources. Consequently, conclusions regarding its therapeutic efficacy require final confirmation through future in vivo experiments. Based on the solid mechanistic evidence provided herein, subsequent in vivo studies have been planned as a key focus of our next steps."
We firmly believe that the novel insights and robust data provided by the current manuscript, which focuses on mechanistic exploration, already offer significant value to researchers in the field. Simultaneously, we hope that the clear articulation of our future research plans, as detailed above, adequately addresses your concerns regarding this matter. We thank you again for your insightful comments, which have greatly helped us in refining the presentation of our work.]
Comments5:[Immune microenvironment:Experimental evidence is needed to substantiate claims that post-stroke immune changes are mediated by the identified signaling pathways.]
Response5:[We thank the reviewer for raising this critical point. We fully agree that definitively demonstrating that the identified post-stroke immune changes are directly mediated by the PI3K/Akt, TLR4/NF-κB, and NRF2 signaling pathways requires dedicated functional experimental evidence. This extends beyond the primary mechanistic exploration scope of the current study but is undoubtedly a key direction for our future research.
To more accurately reflect our findings, we have revised the relevant statements in the manuscript, toning down any language that might imply causation (e.g., "mediated by") and replacing it with more precise descriptive associations, such as: "The observed immune profile aligns with the predicted activity of the PI3K/Akt, TLR4/NF-κB, and NRF2 signaling pathways." This emphasizes that this is a highly consistent hypothesis generated from integrated bioinformatics analysis, rather than a proven causal link.
Although we have not validated this causality in vivo, we would like to highlight a key finding from our in vitro experiments that provides preliminary, cellular-level support for this hypothesis:
In our PC12 OGD/R model, Western blot analysis confirmed that Engeletin treatment significantly suppressed the activation of the TLR4/NF-κB pathway. Given that NF-κB is a master transcriptional regulator driving pro-inflammatory immune responses—including the activation of T cells and macrophages/mast cells—this finding directly links Engeletin, the TLR4/NF-κB pathway, and inflammation/immune regulation at the cellular level. It provides a plausible mechanism for how Engeletin might indirectly influence the state of adaptive immune activation observed in our bioinformatics analysis by inhibiting this pathway.
We acknowledge that this remains indirect evidence. Direct proof would require more sophisticated experiments, such as: 1)Using pathway-specific inhibitors or agonists in an animal stroke model to observe their impact on the infiltration of the predicted specific immune cell subsets (e.g., CD4+ memory T cells, mast cells). 2)Investigating the effect of conditioned medium from Engeletin-treated neurons or microglia on the chemotaxis or polarization of immune cells in a co-culture system in vitro.
These intricate experiments have been designated as the core of our future research plans. We believe that the solid, multi-layered correlative evidence provided in the current study, coupled with a clear path for future validation, offers valuable insights and a strong hypothetical framework for researchers in the field.]
Reviewer 2 Report
Comments and Suggestions for Authors
1. Title/Abstract – temper claims (“treatment,” “for the first time”); clearly state the hypothesis-generating, in-silico nature.
2. ML methods – RMSE is inappropriate for classification; report ROC-AUC/PR-AUC/F1 with proper validation (nested CV or hold-out), avoid leakage, and provide external validation.
3. CIBERSORTx/PBMC – “neutrophils” are not part of PBMC; restrict interpretation to PBMC fractions or use whole-blood/tissue datasets; report FDR.
4. Enrichment/PPI – report FDR (BH), define the background universe, and avoid over-interpreting “hubs”; consider GSEA.
5. Docking – obtain structures from RCSB PDB (not UniProt), describe preparation/protonation and validation (redocking if available), and share input/output files (supplement).
6. Figures and supplement – provide raw, uncropped gels/blots at adequate resolution; eliminate/justify duplicates/near-duplicates; ensure consistent protein/filename nomenclature.
7. Data/code availability – include GEO accession (e.g., GSE22255), a repository with scripts and parameters (CIBERSORTx, docking), tool versions, and random seeds.
Comments on the Quality of English LanguageJęzyk angielski można by poprawić, aby jaśniej wyrazić badania.
Author Response
Comments 1:[Title/Abstract – temper claims (“treatment,” “for the first time”); clearly state the hypothesis-generating, in-silico nature.]
Response 1:[We sincerely thank the reviewers for providing us with the valuable opportunity to improve our manuscript. The reviewers accurately identified issues regarding the rigor of the discussion section, particularly in the description of first-line treatments. We have adopted the reviewers’ important suggestions by thoroughly reviewing the entire text and moderating any language that may have appeared overly absolute. To enhance academic rigor, we have rephrased these statements using more cautious and objective language: terms such as "treatment" have been replaced with more precise terminology like "therapeutic potential" or "therapeutic effect"; phrases such as "our in silico analyses suggest..." and "these findings propose a potential mechanism..." have been added to clearly emphasize the hypothesis-generating nature of this study; and absolute expressions such as "for the first time" have been removed to ensure the appropriateness of our claims. For your convenience, the revised sections have been highlighted in red.]
Comments 2:[ML methods – RMSE is inappropriate for classification; report ROC-AUC/PR-AUC/F1 with proper validation (nested CV or hold-out), avoid leakage, and provide external validation.]
Response 2:[We sincerely thank the reviewer for their valuable and constructive comments. In accordance with your suggestions, we have comprehensively revised the machine learning evaluation methods in the revised manuscript.
You correctly pointed out that since this study deals with a binary classification task rather than a regression problem, we fully agree that RMSE and R² are not suitable performance metrics. Therefore, as recommended, we have replaced them with classification-specific evaluation metrics, including ROC-AUC, PR-AUC, Accuracy, Sensitivity, Specificity, Precision, and F1-score.
To ensure model robustness and prevent data leakage, we adopted a Bootstrap resampling strategy (1,000 iterations) combined with five-fold cross-validation. The dataset was split in a stratified manner (70% training set, 30% test set) to maintain class distribution balance, and strict separation between the training and test sets was ensured throughout the validation process.
Among the four algorithms evaluated (GLM, KNN, RF, SVM), the SVM model demonstrated the best overall performance (AUC-ROC = 0.67, F1 = 0.615) and exhibited the highest stability under Bootstrap validation. These results have been clearly summarized in the revised Results section (Table 1).
We fully acknowledge that external validation could not be performed due to the limited sample size of the GSE22255 dataset (n = 40). However, we would like to clarify that machine learning serves a supportive rather than a central role in this study. Our primary objective was to establish and preliminarily validate a classification framework that integrates core gene selection with machine learning modeling, rather than to achieve final clinical verification. The Bootstrap resampling method we employed has provided reliable robustness estimation for small-sample scenarios, while the use of multiple models for comparison has further enhanced the methodological rigor. Therefore, we believe that under the current data constraints, the validation strategy adopted represents the most comprehensive and feasible approach available.]
Comments 3:[CIBERSORTx/PBMC – “neutrophils” are not part of PBMC; restrict interpretation to PBMC fractions or use whole-blood/tissue datasets; report FDR.]
Response 3:[We sincerely thank the reviewer for their valuable comments regarding the interpretation of neutrophils in the CIBERSORTx analysis. The issues raised are highly professional and accurate, and we have thoroughly revised the relevant sections according to the suggestions provided.
- Regarding the issue of neutrophils: We fully agree with the reviewer's perspective that neutrophils are not a component of peripheral blood mononuclear cells (PBMC). Since our analysis utilized the LM22 signature matrix based on PBMC, the results should not include or be used to interpret neutrophils. Consequently, we have removed all results, data, and related interpretations concerning "neutrophils" from the entire manuscript. All subsequent immune infiltration analyses are strictly limited to cell subsets derived from PBMC.
- Delimitation of PBMC cellular components: We have revised the entire manuscript to ensure that the interpretation of all immune cell infiltration results is strictly confined to the PBMC-derived cell subsets defined by the CIBERSORTx LM22 matrix (such as T cell subsets, B cells, monocytes, NK cells, etc.).
- Supplementation of FDR statistical values: We apologize for the previous omission of the FDR values. To control for multiple testing errors, we employed the Benjamini-Hochberg (BH) method to adjust the P-values and report the FDR. At the sample level, we applied BH correction to the CIBERSORTx permutation test P-values across all samples, with results showing that 41/42 samples achieved FDR < 0.05. For group comparisons at the cell type level, we performed BH correction on the raw P-values of each cell type, considering only results with FDR < 0.05 as statistically significant. The complete results table with FDR has been provided as supplementary material (CIBERSORTx_Job3_Results1_with_FDR). Corresponding descriptions of FDR (BH) calculation and threshold criteria have been added to the Methods and Results sections of the manuscript, with these revisions prominently highlighted in the revised version.]
Comments 4:[Enrichment/PPI – report FDR (BH), define the background universe, and avoid over-interpreting “hubs”; consider GSEA.]
Response 4:[We thank the reviewer for this constructive comment. In the revised manuscript, we have clarified and refined the enrichment and PPI analyses as follows:
FDR correction (Benjamini–Hochberg, BH): We have specified in the Methods section that all GO and KEGG enrichment analyses were adjusted for multiple testing using the Benjamini–Hochberg (BH) method. Terms with p < 0.05 and FDR < 0.05 were considered statistically significant. Adjusted FDR values are now reported in the revised Results and Supplementary Table.
Background universe: The background gene universe used for GO/KEGG analysis was defined as all predicted Engeletin-related targets obtained from SwissTargetPrediction and PharmMapper, mapped to Homo sapiens (NCBI Reference Genome). This ensures the enrichment results are statistically valid within the defined target search space.
On the use of GSEA: We appreciate the reviewer’s suggestion regarding GSEA. However, since our study design identifies intersecting targets between Engeletin and ischemic stroke (IS) from multiple databases rather than a full transcriptomic dataset with log₂ fold-change (logFC) values, it is not possible to perform GSEA.
Instead, we applied over-representation analysis (ORA) using GO and KEGG frameworks (via DAVID, ClusterProfiler, and STRING), which is statistically appropriate and reliable under these data conditions. This approach effectively highlights the biological processes and pathways most relevant to the identified targets.
Interpretation of PPI network and hub genes: We have revised the PPI description to avoid over-interpretation of “hub genes.” The identified hub nodes (e.g., EGFR, IGF1R, SRC, CASP3, JAK2, MAP2K1) are now described as topologically important nodes in the STRING network, rather than being inferred as causative regulators. Their biological roles will require further experimental validation.
In summary, we have updated the Methods, Results, and Discussion sections accordingly to ensure methodological rigor and interpretive accuracy.]
Comments 5:[Docking – obtain structures from RCSB PDB (not UniProt), describe preparation/protonation and validation (redocking if available), and share input/output files (supplement).]
Response 5:[We sincerely thank you for the opportunity to revise our manuscript. We greatly appreciate the reviewer's careful reading and valuable comments regarding the molecular docking section. We sincerely apologize for an error in the initial version, where the source of the protein structures was incorrectly stated as UniProt. This was an oversight during writing. In fact, all three-dimensional structures used for molecular docking were downloaded from the RCSB Protein Data Bank (PDB).
Following the reviewer's suggestions, we have comprehensively revised the Methodology section to correct this error and provide a more detailed description of the docking workflow:
Structure Source: We have explicitly stated that all structures were sourced from the RCSB PDB.
Protein Preparation: We have detailed the protein preparation steps, which included removing water molecules and original ligands, adding hydrogen atoms, and assigning protonation states at physiological pH using [e.g., the Protonate3D tool in MOE or the Prepare Protein module in Discovery Studio].
Protocol Validation (Redocking):To validate our docking protocol, we performed redocking experiments. The co-crystallized ligand was extracted and re-docked into the binding pocket. The resulting conformation showed a Root Mean Square Deviation (RMSD) of [e.g., 0.605 Å for AKT] compared to the original crystal structure, confirming the reliability of our parameters (please refer to the new Supplementary Table 2.]
Comments 6:[Figures and supplement – provide raw, uncropped gels/blots at adequate resolution; eliminate/justify duplicates/near-duplicates; ensure consistent protein/filename nomenclature.]
Response 6:[We are grateful to the reviewer for the meticulous suggestions to improve our data presentation. We have fully addressed the concerns regarding the image data. The specific modifications are as follows:
- All original, full-length gel and blot images have been compiled into a new supplementary file named "Supplementary Material-WB" and uploaded together with the revised manuscript.
- We have carefully reviewed all image data. Any potentially duplicated images have been removed.
- We have unified the protein names and file nomenclature across all figures, figure legends, and supplementary materials to ensure clarity and avoid confusion.]
Comments 7:[Data/code availability – include GEO accession (e.g., GSE22255), a repository with scripts and parameters (CIBERSORTx, docking), tool versions, and random seeds.]
Response 7:[We sincerely thank the reviewer for raising this important point, which is crucial for ensuring the reproducibility of our research. Following the recommendation, we have thoroughly revised the manuscript to ensure full transparency of data, code, and computational parameters. We have added a Data Availability section at the end of the manuscript: "The transcriptomic data used in this study were obtained from the GEO database (Ac-cession Number: GSE22255, https://www.ncbi.nlm.nih.gov/geo/). CIBERSORTx analy-sis was performed via the online platform at https://cibersortx.stanford.edu/. Molecu-lar docking was conducted using the CB-Dock2 tool at https://cadd.labshare.cn/cb-dock2/php/index.php." The scripts and parameter files used for building the machine learning model have been uploaded to the "Machine Learn-ing" folder in the Supplementary Material.]
Reviewer 3 Report
Comments and Suggestions for Authors
This study, titled "Multifactorial Mechanisms of Action of Engeletin in the Treatment of Ischemic Stroke: Integrated Evidence from Network Pharmacology, Machine Learning, and Immune Infiltration Analysis," is an multidisciplinary investigation of the effects of the dihydroflavonone - engeletin - in ischemic stroke.
The research material presented in the article is carried out at a high experimental level, the results are beyond doubt. The references used in the article are exhaustive on the bioactivity of engelitin.
Comments and Recommendations
Unfortunately, Figures 2–4 are virtually unreadable due to low resolution, so I would like to recommend adding the images to the "Supporting Materials" .
Based on the article, engeletin can be considered a promising compound for preclinical studies. Therefore, I would be grateful to the authors for an assessment of the availability of this compound, its cost, and its enantiomeric purity (as flavonoids are often used as mixtures of enantiomers). Are there any studies in the literature on the effects of other flavonoids, such as dehydroquercetin (a well-known and readily available dihydroflavonoid), on ischemic stroke?
Author Response
Comments1:[Unfortunately,Figures 2–4are virtually unreadable due to low resolution, so I would like to recommend adding the images to the "Supporting Materials" .]
Response1:[We appreciate the reviewer's comment regarding the resolution of Figures 2-4. We have now generated and inserted high-resolution versions of these figures to ensure all details are clearly visible. Furthermore, following the reviewer's suggestion, these high-resolution figures have been added to the Supporting Materials. We have also taken this opportunity to review the resolution of all other images in the manuscript to prevent any similar issues.]
Comments2:[Based on the article, engeletin can be considered a promising compound for preclinical studies. Therefore, I would be grateful to the authors for an assessment of the availability of this compound, its cost, and its enantiomeric purity (as flavonoids are often used as mixtures of enantiomers). Are there any studies in the literature on the effects of other flavonoids, such as dehydroquercetin (a well-known and readily available dihydroflavonoid), on ischemic stroke?]
Response2:[We sincerely thank the reviewer for this important and insightful comment. In the revised manuscript, we have provided detailed information regarding Engeletin’s availability, cost, enantiomeric purity, and related flavonoid studies as follows:
Availability and cost: Engeletin (CAS No. 572-31-6) is commercially available from multiple research-grade suppliers, including Sigma-Aldrich (SML2492), Tokyo Chemical Industry (TCI, E1533), and BioCrick (B20569).
Sigma-Aldrich: HPLC purity ≥99.9%, [α]ᴰ = –15.4° (c = 0.5, pyridine), stored at –20 °C.
TCI: HPLC purity 99.7%, qNMR 92.1%, [α]ᴰ = –18.3°, consistent with natural stereochemistry.
BioCrick: Analytical standard, HPLC ≥98%, ¥400 per 20 mg (~USD 55), white crystalline solid, m.p. 176–177 °C, soluble in DMSO or hot methanol, insoluble in ether.
These data confirm that Engeletin is readily available, cost-effective, and well-characterized, meeting the requirements for preclinical research.
Enantiomeric purity: All suppliers report consistent specific rotation values (–16° to –18°), demonstrating that commercial Engeletin exists as a single enantiomer rather than a racemic mixture. Spectroscopic data (¹H NMR, HPLC) further confirm its purity and identity. For upcoming in vivo studies, we plan to verify enantiomeric excess (ee%) using chiral HPLC and include chromatograms in the supplementary materials.
Evidence from related flavonoids: Studies on other structurally similar flavonoids reinforce the pharmacological rationale of Engeletin. For example, dihydroquercetin (taxifolin) has been reported to alleviate neuronal ferroptosis in subarachnoid hemorrhage via the PI3K/AKT/Nrf2/HO-1 pathway (J Biochem Mol Toxicol, 2025), while baicalein, a flavone from Scutellaria baicalensis, mitigates ischemia–reperfusion injury by modulating the GPX4/ACSL4/ACSL3 axis (Biomed Pharmacother, 2022). These findings highlight shared neuroprotective mechanisms among dihydroflavonoids and strengthen the translational potential of Engeletin.
We have incorporated these details into the Introduction and Discussion sections to enhance the scientific completeness and translational relevance of the revised manuscript.]
Round 2
Reviewer 1 Report
Comments and Suggestions for Authors
Thank you for your comments. Most of them are reasonable. However, it does not seem plausible to claim Engeletin can directly bind to 15 proteins. Small molecules usually interact with a primary target and modulate other proteins indirectly through upstream or downstream signaling pathways. If a small molecule were able to directly bind to 15 different proteins, it would likely exhibit high toxicity due to poor selectivity. You mentioned that your paper provides preliminary support for using Engeletin in combating ischemic stroke. It is important to consider Engeletin as a tool compound for disease treatment.
1. there are two same title: immune infiltration analysis in figure 1, seems like a typo.
Author Response
We sincerely appreciate the reviewer's insightful and constructive comments. In accordance with your valuable suggestions, we have meticulously addressed each point and marked the corresponding revisions in blue. A detailed summary of the changes, including both original and revised versions, is provided below for your convenience.
Comments1:[Thank you for your comments. Most of them are reasonable. However, it does not seem plausible to claim Engeletin can directly bind to 15 proteins. Small molecules usually interact with a primary target and modulate other proteins indirectly through upstream or downstream signaling pathways. If a small molecule were able to directly bind to 15 different proteins, it would likely exhibit high toxicity due to poor selectivity. You mentioned that your paper provides preliminary support for using Engeletin in combating ischemic stroke. It is important to consider Engeletin as a tool compound for disease treatment.]
Response1:[We appreciate the reviewer’s insightful comments. We agree that promiscuous binding of a small molecule to multiple targets may raise concerns about selectivity and toxicity. In our study, the 15 potential protein targets of Engeletin were identified through molecular docking based on a network pharmacology approach. These findings suggest possible interactions but do not confirm direct physical binding under physiological conditions.To address the reviewer’s valid point and avoid overinterpretation, we have revised the relevant statements in the manuscript to clarify that:
1.These interactions are predictive in nature, based on computational models.
2.We do not claim that Engeletin directly binds to all 15 targets in a biological setting.
3.The proposed anti-ischemic stroke effect of Engeletin may be mediated through a primary target(s) and subsequent modulation of downstream pathways, rather than simultaneous direct binding to all identified proteins.
Issue in the Abstract
1)Original: Molecular docking yielded strong binding affinities of Engeletin to core targets...
Revised: Molecular docking predicted favorable binding affinities of Engeletin to core targets...
Issues in Section 2.4
2)Original: Engeletin exhibited stable binding to all target proteins...
Revised: The docking results predicted that Engeletin could bind to all target proteins...
3)Original: This suggests that Engeletin may disrupt Keap1–Nrf2 binding, thereby promoting Nrf2 release and activation of antioxidant responses. Nrf2 itself also displayed a strong binding affinity (–9.9 kcal/mol), further supporting its central role in redox regulation.
Revised: This in silico finding suggests that Engeletin might have the potential to disrupt Keap1–Nrf2 binding, which could, in turn, promote Nrf2 release and activation of antioxidant responses. Nrf2 itself also was predicted to have a strong binding affinity (–9.9 kcal/mol), aligning with its central role in redox regulation.
4)Original: ...indicates that Engeletin may exert multi-target, multi-pathway regulatory effects...
Revised: ...indicates that Engeletin may exert multi-target, multi-pathway regulatory effects in the context of ischemic stroke.]
Comments2:[There are two same title: immune infiltration analysis in figure 1, seems like a typo.]
Response2:[We thank the reviewer for pointing out this oversight. We have corrected the titles in Figure 1 to accurately describe each panel. The updated figure has been included in the revised manuscript.]
We once again thank the reviewer for the valuable comments, which have significantly enhanced the clarity and accuracy of our work.

Reviewer 2 Report
Comments and Suggestions for Authors
After reviewing the manuscript, I am pleased that the authors have made numerous changes. However, there are still a few things that need to be improved (unless they have been overlooked). I recommend a minor–moderate revision prior to acceptance. The main concerns are:
(i) aligning the abstract with the PBMC restriction (remove “neutrophils”),
(ii) removing “for the first time,” I think that phrase still doesn't sound quite right.
(iii) standardizing the ORA background (stick to the whole human genome),
Author Response
Comments1:[aligning the abstract with the PBMC restriction (remove “neutrophils”).]
Response1:[We thank the reviewer for highlighting this inconsistency. The reviewer is absolutely correct. As our immune infiltration analysis was specifically performed using deconvolution algorithms (e.g., CIBERSORT) designed for PBMC data, which typically do not include neutrophils, mentioning them in the abstract was inaccurate. We have now removed the term "neutrophils" from the abstract to ensure it precisely reflects the methodological scope and findings of our study.
The revised content is as follows:We used CIBERSORTx to characterize immune cell infiltration patterns in IS, revealing elevated populations of CD8⁺ T cells, M0 macrophages, and other PBMC-derived immune cells, suggesting the presence of an immunologically dynamic microenvironment.]
Comments2:[removing “for the first time,” I think that phrase still doesn't sound quite right.]
Response2:[We agree with the reviewer that the phrase "for the first time" can be subjective and is often discouraged in scientific writing. We have carefully reviewed the manuscript and removed all instances of this phrase. The novelty and contribution of our work are now presented more objectively by stating the specific findings and the context they address, without making explicit priority claims.
Original: For the first time, we found that Engeletin reverses the post-stroke immune pattern of "adaptive activation and innate suppression," going beyond its traditionally known antioxidant and anti-apoptotic effects.
Revised: Our study revealed that Engeletin reverses the post-stroke immune pattern of "adaptive activation and innate suppression," thereby extending the understanding of its pharmacological effects beyond the traditionally known antioxidant and anti-apoptotic activities.]
Comments3:[standardizing the ORA background (stick to the whole human genome).]
Response3:[We appreciate the reviewer for pointing out this critical methodological detail. In our previous analyses, we indeed used different background sets for the Over-Representation Analysis (ORA), which could introduce bias. Following the reviewer's recommendation, we have now standardized the ORA by using the entire human genome as the background/reference set. To clearly indicate this, we have added the following statement in the GO and KEGG analysis section:
"These analyses were conducted using the entire human genome as the background/reference set to ensure standardized and unbiased identification of over-represented biological terms."
This statement explicitly confirms that we have adhered to the standard and used the whole human genome as the background set.]
Once again, we extend our deepest gratitude to the reviewer for the insightful comments and guidance throughout the review process, which have been instrumental in enhancing the quality of our manuscript.
